# The Latent Guardian: Defending Collaborative Perception via Feature-Level Consistency Verification

Zhuangzhuang Zhang [1]   Mingxin Li [1]   Libing Wu [2]   Wei-Bin Lee [3]   Jianping Wang [1]

## Abstract

Collaborative perception (CP) significantly extends the sensing range of connected and autonomous vehicles (CAVs). However, its reliance on data fusion among multiple CAVs makes it inherently vulnerable to adversarial attacks from malicious participants. Existing defenses primarily rely on output-level consensus, assuming that malicious messages manifest as statistical outliers, while suffering from poor adaptability to environmental noise. This makes them vulnerable to stealthy adversarial attacks and prone to high false positive rates. To address this challenge, we shift the defense paradigm from superficial output-level consensus to deeper consistency within the internal feature space. Guided by this principle, we propose `Cerberus`, a novel defense framework against adversarial attacks in CP systems by leveraging multi-dimensional consistency in the feature space. By quantifying conflicts in topological structure, semantic direction, and energy distribution within feature maps, `Cerberus` effectively detects adversarial perturbations and provides dynamic protection against adversarial attacks. Experimental results demonstrate that `Cerberus` significantly outperforms state-of-the-art methods, effectively limiting the attack success rate to as low as 0.05% while restoring the AP to 0.88.

## 1. Introduction

Collaborative perception (CP) has emerged as a transformative paradigm for autonomous driving, enabling the ego vehicle to see beyond its physical line-of-sight by sharing sensory information with neighboring connected and autonomous vehicles (CAVs) (Dao et al., 2024; Gao et al., 2024; Lei et al., 2022). This capability effectively overcomes the inherent occlusion and long-range sensing limitations of single-vehicle perception systems (Huang et al., 2025; Hong et al., 2024; Tang et al., 2025). Technically, CP systems can be categorized based on the stage of data sharing and fusion into three types: early fusion (raw sensor data), intermediate fusion (intermediate feature maps), and late fusion (detection results). Among these, intermediate fusion has become the dominant paradigm in state-of-the-art CP frameworks (Xu et al., 2022), as it achieves an optimal trade-off between perceptual accuracy and communication overhead.

However, this open information exchange introduces a critical vulnerability, leaving the system susceptible to adversarial attacks (Lin et al., 2025; Li et al., 2024). Malicious vehicles can broadcast fabricated or perturbed feature maps, inducing phantom objects or masking real hazards, which can lead the ego vehicle into catastrophic traffic accidents (Tu et al., 2021; Yang et al., 2023). To mitigate this problem, numerous studies have been proposed. These existing defense mechanisms can generally be classified into two categories as classifier-based (Hu et al., 2025a; Tao et al., 2025) and consensus-based methods (Hu et al., 2025b; Li et al., 2023b; Zhang et al., 2024; Zhao et al., 2024). Despite these efforts, these approaches still suffer from inherent limitations.

Classifier-based approaches typically involve training a binary discriminator on large-scale labeled datasets to distinguish malicious messages from benign ones. This causes them to require extensive labeled data and struggle to generalize to zero-day attacks in open-world CP environments, leading to their impracticality for diverse real-world deployment. Driven by these limitations, recent research has shifted its focus toward consensus-based methods, which are inherently unsupervised and more flexible. Nonetheless, their effectiveness relies on *two key assumptions*. (1) **Discernible output divergence**: malicious outputs (e.g., bounding boxes or object scores) exhibit significant and detectable deviations from those of benign CAVs. (2) **Static environ-**

---

[1]Department of Computer Science, City University of Hong Kong, Hong Kong, China [2]School of Cyber Science and Engineering, Wuhan University, Wuhan, China [3]Hon Hai Research Institute, Taipei, Taiwan. Correspondence to: Libing Wu <wu@whu.edu.cn>, Jianping Wang <jianwang@cityu.edu.hk>.

*Proceedings of the 43rd International Conference on Machine Learning*, Seoul, South Korea. PMLR 306, 2026. Copyright 2026 by the author(s).

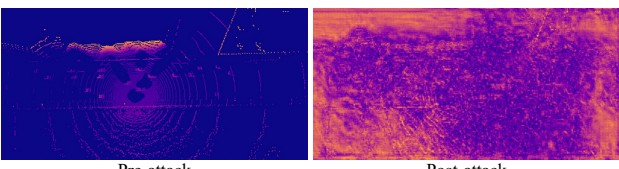

Pre-attack    Post-attack

*Figure 1.* Visualization of feature maps from the attacker vehicle before and after adversarial perturbation. The clean representation (left) exhibits distinct spatial structures and semantic patterns consistent with the scene. In contrast, the attacked representation (right) is overwhelmed by chaotic noise, resulting in a complete loss of topological integrity. This result demonstrates that the attack not only disturbs activation magnitudes but fundamentally disrupts the structural consistency of the feature representations.

*mental conditions*: environmental noise follows a stable, predictable distribution, enabling the use of a fixed decision boundary.

These assumptions severely limit the practicality of consensus-based defenses. On one hand, adversarial perturbations can be engineered to be stealthy enough to evade detection in the output domain. On the other hand, environmental factors like adverse weather introduce stochastic noise that rigid thresholds fail to distinguish from actual attacks, leading to high false-alarm rates. To address these issues, we propose a paradigm shift from surface-level output consensus to intrinsic consistency at the internal feature level. Our key insight is that intermediate feature representations are not arbitrary tensors, but structured projections of physical scenes that must reside on a shared latent manifold induced by the real world. As illustrated in Figure 1, features derived from real-world objects exhibit strong intrinsic regularities, manifested as coherent spatial organization, stable semantic activation patterns, and well-behaved energy responses. In contrast, adversarial attacks introduce artificial perturbations that directly manipulate intermediate representations, leading to off-manifold deviations that violate these intrinsic constraints, even when the resulting outputs appear plausible. Therefore, effective defense should move beyond output-level consensus and examine the physical plausibility and manifold consistency of shared features.

Building on this insight, we propose `Cerberus`, a novel defense framework *acting as a **Latent Guardian** for CP* systems. `Cerberus` enforces multi-dimensional consistency over the feature maps shared by each CAV across three complementary dimensions, enabling the detection of stealthy adversarial attacks. Specifically, we first employ an *Isomorphic View Fusion* module that constructs two verification groups with equivalent fields of view via interleaved sampling, effectively deconfounding geometric viewpoint misalignment. Subsequently, our *Feature-Level Consistency Verification* mechanism examines deep representations by quantifying conflicts between these two verification groups and the ego vehicle's features across three dimensions: topo-

logical integrity, semantic direction, and energy distribution. Finally, to ensure robustness against environmental noise, we design an *Uncertainty-Aware Adaptive Decision* module that leverages ego-feature entropy to dynamically calibrate decision boundaries. Our main contributions are as follows:

- We propose `Cerberus`, a novel defense framework that effectively detects stealthy adversarial attacks by enforcing multi-dimensional consistency within the deep feature space.

- We design an isomorphic view fusion module that eliminates geometric viewpoint bias via interleaved sampling. Based on this unbiased reference, we design a multi-dimensional metric that strictly enforces topological integrity, semantic direction, and energy distribution constraints to detect adversarial attacks.

- We design an uncertainty-aware adaptive decision module that integrates perceptual entropy to quantify environmental ambiguity. By dynamically calibrating decision boundaries according to real-time uncertainty, our framework strikes an optimal balance between security and availability.

## 2. Related Work

**Consensus-based defenses.** This class of defense methods (Hu et al., 2025b; Li et al., 2023b; Zhang et al., 2024; Zhao et al., 2024) utilizes a sampling mechanism to continuously verify the consistency of the perception results from various collaborators, thereby filtering out and fusing the data only from those collaborators that exhibit perceptual consensus. For example, in ROBOSAC (Li et al., 2023b), a subset of collaborating vehicles is randomly selected for consensus validation to facilitate the identification of malicious CAVs. Although CP-Guard (Hu et al., 2025b) does not require prior knowledge of the attacker's capabilities, it employs a fixed detection threshold, which renders it vulnerable to stealthy attacks. Our framework is also a consensus-based defense, but unlike existing methods, it requires no prior knowledge of attackers and innovatively leverages correspondences from the physical to the feature level to effectively defend against stealthy attacks.

**Classifier-based defenses.** In contrast to the previously discussed consensus-based defense methods, classifier-based approaches (Hu et al., 2025a; Tao et al., 2025) typically identify attackers by training a specialized classifier. This characteristic necessitates the use of extensive labeled datasets and results in poor generalization capability in unseen scenarios (e.g., those not present in the training data). For instance, GCP (Tao et al., 2025) requires training an LSTM-based classifier to identify malicious vehicles interspersed among

benign CAVs. However, the training of such an LSTM neural network necessitates the prior collection of attack datasets, which is inherently difficult and results in high deployment costs.

## 3. System and Threat Model

### 3.1. System Model

Similar to (Wang et al., 2025; Zhang et al., 2024; Hu et al., 2025b), we consider a CP system, in which each CAV collects perception data from its onboard sensors, extracts intermediate features, and then transmits these features to neighboring CAVs via wireless communication. After receiving the intermediate feature maps, each CAV fuses them using an intermediate fusion algorithm such as AttFusion (Xu et al., 2022), thereby obtaining complete perception results that integrate information from multiple CAVs. The fusion process can be formally described as follows:

Let $\mathcal{N}_i$ denote the set of neighboring CAVs of vehicle $i$ (including itself), and let $\mathbf{F}_j \in \mathbb{R}^{C \times H \times W}$ be the intermediate feature map extracted by $\text{CAV}_j$. The fused feature map at $\text{CAV}_i$ is obtained by

$$\mathbf{F}_i = \mathcal{F}_{\text{fusion}} \left( \{\mathbf{F}_j\}_{j \in \mathcal{N}_i} \mid \Theta \right),$$

where $\mathcal{F}_{\text{fusion}}(\cdot \mid \Theta)$ represents the intermediate fusion module with learnable parameters $\Theta$.

### 3.2. Threat Model

In this paper, we primarily consider the insider attack threats introduced by CP. Compared to traditional adversarial attacks targeting single-vehicle perception, this represents a novel attack paradigm (Zhang et al., 2024). Therefore, we are committed to addressing the security threats posed by this emerging class of attacks. Specifically, following existing insider attack models (Tu et al., 2021; Hu et al., 2025b; Zhang et al., 2024; Wang et al., 2025) against CP systems, we assume that the attacker gains control of a single CAV. Through adversarial optimization techniques, the attacker carefully crafts malicious feature maps and transmits them to the ego vehicle (i.e., the victim vehicle), thereby compromising the ego vehicle's CP fusion results. Consistent with previous work (Hu et al., 2025b; Zhang et al., 2024; Wang et al., 2025) in this domain, we consider white-box attacks, where the attacker has access to the victim vehicle's model architecture and weight parameters.

## 4. Defense Framework

### 4.1. Framework Overview

As mentioned above, existing defenses for CP (Li et al., 2023b; Zhang et al., 2024) primarily rely on statistical consistency at the output level (Wang et al., 2025). However,

this paradigm is vulnerable to adversarial attacks, as attackers can evade detection by introducing slight feature perturbations that produce visually plausible outputs. To address this gap, we propose Cerberus, a novel feature consistency-based defense framework. Unlike traditional methods (Li et al., 2023b; Hu et al., 2025b), Cerberus no longer relies solely on simple validation of output-layer consensus. Our key insight is that while attackers can easily forge perceptual outputs, as illustrated in Figure 1, they cannot generate feature maps that closely resemble those of benign vehicles.

As shown in Figure 2, Cerberus comprises three key components. First, the *Isomorphic View Fusion* module constructs verification groups with overlapping fields of view via angular ranking and interleaved sampling, mitigating viewpoint misalignment. Second, the *Feature-Level Consistency Verification* module detects attacks by quantifying deep feature conflicts between each verification group and the ego vehicle across three dimensions: topological integrity, semantic direction, and energy distribution. Finally, the *Uncertainty-Aware Adaptive Decision* module leverages feature entropy to dynamically calibrate defense thresholds, ensuring robustness against environmental noise.

### 4.2. Isomorphic View Fusion

To reliably detect malicious vehicles, a key challenge in CP is distinguishing feature inconsistencies caused by benign viewpoint differences from those induced by adversarial attacks. Due to heterogeneous viewpoints, vehicles naturally observe different parts of the scene, making such inconsistencies inherently ambiguous from the ego vehicle's perspective. To address this issue, we propose an isomorphic viewpoint grouping strategy that partitions vehicles into two groups with similar fields of view based on their angular positions. By aggregating perceptions within each group, viewpoint-induced discrepancies are normalized, allowing residual inconsistencies between the two fused views to be more reliably attributed to adversarial behavior. In contrast to existing sampling-based methods (Hu et al., 2025b; Li et al., 2023b), which overlook viewpoint geometry and thus suffer from high false-positive rates, our approach explicitly accounts for viewpoint heterogeneity.

Specifically, the process begins by quantifying the spatial relationship between the ego vehicle and its neighbors. Let $\mathcal{N} = \{n_1, n_2, \dots, n_N\}$ denote the set of $N$ collaborative neighbors. For each neighbor $n_i$, we calculate its relative azimuth angle $\theta_i \in [-\pi, \pi]$ with respect to the ego vehicle's heading. To ensure spatial continuity, we sort the neighbors based on their angular deviation:

$$\mathcal{N}_{sorted} = \text{argsort}(\{\theta_1, \theta_2, \dots, \theta_N\}). \qquad (1)$$

This sorting step effectively linearizes the angular distribu-

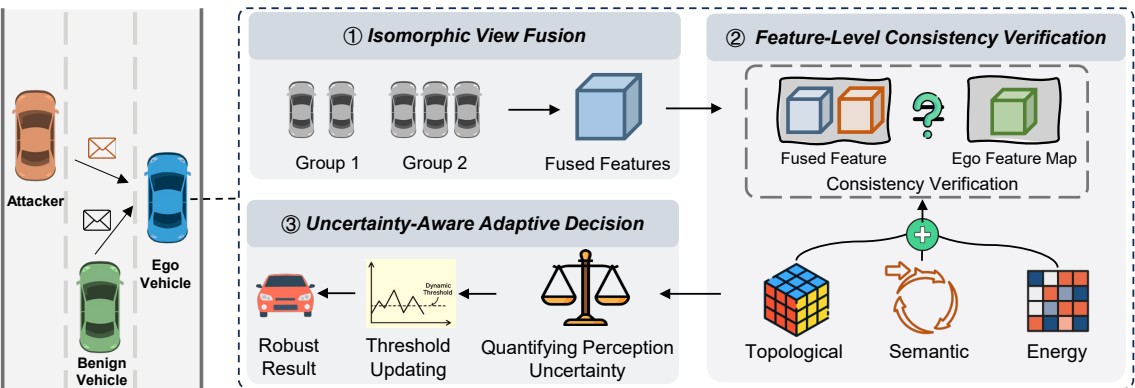

*Figure 2.* Overview of `Cerberus`. The `Cerberus` defense attacks through three steps: (1) Constructing geometrically isomorphic verification groups via ***Isomorphic View Fusion*** to remove viewpoint bias; (2) Detecting deep feature inconsistencies using ***Feature-Level Consistency Verification*** metrics; and (3) Performing robust decision-making with ***Uncertainty-Aware Adaptive Decision***, which dynamically adjusts threshold based on environmental uncertainty, thereby yielding robust fused perception results.

tion of the surrounding traffic, clustering spatially adjacent neighbors in the sorting index sequence.

Building upon this sorted sequence, we proceed to the grouping phase via an interleaved assignment strategy. Specifically, neighbors in the sorted sequence $\mathcal{N}_{sorted}$ are alternately assigned to two disjoint subsets, $\mathbb{G}_1$ and $\mathbb{G}_2$: $\mathbb{G}_1 = \{\mathcal{N}_{sorted}[2k]\}$, $\mathbb{G}_2 = \{\mathcal{N}_{sorted}[2k+1]\}$. This stratified sampling mechanism ensures that both groups maintain a uniform angular density across the entire $360°$ field of view. Consequently, $\mathbb{G}_1$ and $\mathbb{G}_2$ become isomorphic observers, minimizing the baseline geometric discrepancy in the absence of attacks.

Following the grouping, we align the feature maps from all neighbors to the ego's coordinate system using a differentiable warping function $\mathcal{W}(\cdot)$. The features within each isomorphic group are then aggregated to synthesize the group-level representation:

$$\mathbf{F}_{\mathbb{G}_k} = \mathcal{F}_{\text{fusion}\,i \in \mathcal{G}_k}(\mathcal{W}(\mathbf{F}_i, \xi_i \to \xi_{ego})) \qquad (2)$$

where $\mathcal{F}_{\text{fusion}}$ is a permutation-invariant aggregation function, and $\xi$ represents the pose. This yields two fused feature maps, $\mathbf{F}_{\mathbb{G}_1}$ and $\mathbf{F}_{\mathbb{G}_2}$, which serve as the unbiased geometric references for the subsequent *feature-level consistency verification* module.

### 4.3. Feature-Level Consistency Verification

After performing isomorphic view fusion on CAVs, we design the *Feature-Level Consistency Verification* module to intercept adversarial attacks at their root. Distinct from existing defenses that operate at the post-decoding output level, our approach leverages the rich semantics of the high-dimensional latent space. While prior methods suffer from irreversible information loss due to quantization and non-maximum suppression, our framework adheres to a con-

tinuous latent manifold governed by the scene's physical topology. This enables us to rigorously inspect consistency across topology, semantics, and energy, effectively intercepting attacks at their root.

**Topological integrity.** Natural scene features exhibit strong local spatial correlations and continuous topological structures (e.g., the continuous edges of roads or vehicles). In contrast, adversarial perturbations often manifest as high-frequency noise or disjoint patches that disrupt this local continuity. We utilize the feature-level structural similarity (SSIM) (Wang et al., 2004) to quantify this topological consistency. Formally, given the group feature $F_{\mathbb{G}_k}$ and ego feature $F_{ego}$, the structural similarity is calculated using a sliding window approach:

$$S(\mathbf{F}_{\mathbb{G}_k}, \mathbf{F}_{ego}) = \frac{(2\mu_k \mu_{ego} + \rho_1)(2\sigma_{k,ego} + \rho_2)}{(\mu_k^2 + \mu_{ego}^2 + \rho_1)(\sigma_k^2 + \sigma_{ego}^2 + \rho_2)},$$
$$(3)$$

where $\mu_k$ and $\mu_{ego}$ represent the local mean, $\sigma_k^2$ and $\sigma_{ego}^2$ denote the local variance, and $\sigma_{k,ego}$ is the covariance. $\rho_1$ and $\rho_2$ are small constants for numerical stability. Finally, we define the topological discrepancy score as the deviation from perfect similarity:

$$D_{topo} = 1 - S(\mathbf{F}_{\mathbb{G}_k}, \mathbf{F}_{ego}). \qquad (4)$$

A high $D_{topo}$ implies that the candidate feature violates the spatial dependencies observed by the ego, indicating a structurally incoherent fabrication.

**Semantic direction.** The direction of a feature vector in the high-dimensional latent space encodes its semantic intent (e.g., pushing the decision boundary towards the vehicle class). Adversarial attacks often attempt to inject false semantics by forcing the feature vector to rotate away from the ground-truth direction. Instead of computationally expensive gradient estimations, we directly verify this semantic

alignment using cosine dissimilarity:

$$D_{sem} = 1 - \frac{\mathbf{F}_{\mathbb{G}_k} \cdot \mathbf{F}_{ego}}{\|\mathbf{F}_{\mathbb{G}_k}\|_2 \|\mathbf{F}_{ego}\|_2}, \qquad (5)$$

where $\cdot$ denotes the dot product, $k \in \{1, 2\}$, and $\| \cdot \|_2$ represents the $L_2$ norm. In this manner, we effectively expose the semantic conflict between malicious CAVs and the ego vehicle.

**Energy distribution shift.** Adversarial attacks typically introduce high-frequency noise or abnormal activation magnitudes to deceive the network, resulting in an unnatural energy distribution compared to benign features. We measure this signal distortion via the activation magnitude discrepancy ($D_{mag}$), defined as a non-linear mapping of the $L_1$ distance:

$$D_{mag} = \exp\left(\frac{\|\mathbf{F}_{ego} - \mathbf{F}_{\mathbb{G}_k}\|_1}{\sigma}\right), \qquad (6)$$

where $\sigma$ is a scaling factor to normalize the sensitivity. A sharp increase in $D_{mag}$ indicates the presence of significant adversarial perturbations.

**Joint anomaly score.** To provide a unified metric for the subsequent decision module, we aggregate the values from the three dimensions of topological integrity, semantic direction, and energy distribution shift, thereby obtaining a comprehensive joint anomaly score. The score $J_k$ for the group $\mathbb{G}_k$ is formulated as:

$$J_k = \lambda_1 D_{topo} + \lambda_2 D_{sem} + \lambda_3 D_{mag}, \qquad (7)$$

where $\lambda_{1,2,3}$ are balancing coefficients satisfying $\sum \lambda_i = 1$. A higher $J_k$ indicates a higher probability that the collaborator group contains malicious CAVs, marking it as a potential threat to the CP.

### 4.4. Uncertainty-Aware Adaptive Decision

In this step, our primary goal is to distinguish between malicious and benign CAVs using the anomaly score obtained from the previous stage. This requires establishing a clear detection threshold to classify scores as anomalous or legitimate. However, in complex collaborative environments (e.g., adverse weather or severe occlusion), the ego vehicle's perception capability is inevitably degraded. In such scenarios, relying on a static detection threshold is highly prone to false positives, as legitimate discrepancies caused by environmental noise may be mistakenly classified as malicious perturbations. In other words, degraded ego perception amplifies the feature variance across collaborators, rendering any fixed threshold either too lenient (missing attacks) or too strict (triggering false alarms). To address this, we propose an *Uncertainty-Aware Adaptive Decision* mechanism that dynamically calibrates the decision boundary based on both historical statistics and real-time ego uncertainty.

**Ego-uncertainty quantification.** First, to gauge the reliability of the current observation, we measure the ego vehicle's perception uncertainty via feature entropy. High entropy indicates low confidence in the ego's observation. A relatively uniform feature distribution suggests that the ego vehicle cannot reliably discriminate between scene elements under challenging conditions such as occlusion or adverse weather. The normalized uncertainty $U_t \in [0, 1]$ for frame $t$ is computed as:

$$U_t = \frac{-1}{\log(C)} \sum_{c=1}^{C} p(f_{ego}^c) \log p(f_{ego}^c), \qquad (8)$$

where $f_{ego}^c \in \mathbb{R}^{H \times W}$ denotes the $c$-th channel of the ego feature map, $C$ is the number of channels, and $H$ and $W$ are the spatial height and width, respectively. The channel-wise probability is given by

$$p(f_{ego}^c) = \frac{1}{HW} \sum_{h,w} \frac{\exp(f_{ego}^c(h, w))}{\sum_{c'=1}^{C} \exp(f_{ego}^{c'}(h, w))}. \qquad (9)$$

**Threshold computation.** Instead of a fixed baseline, we maintain a sliding window of historical anomaly scores from all benign CAVs. By maintaining this sliding window, we approximate the distribution of anomaly scores for current benign CAVs. Under the assumption that benign gradient variations follow a Gaussian distribution due to the Central Limit Theorem applied to stochastic gradient descent dynamics (Mandt et al., 2017), we can mathematically define the upper bound of normal behavior. The statistical limit $T_{stat}$ at frame $t$ is derived using the 3-Sigma rule:

$$T_{stat} = \mu_t + \gamma \cdot \sigma_t, \qquad (10)$$

where $\mu_t$ and $\sigma_t$ represent the moving average and standard deviation of the benign window, and $\gamma = 3$ is the sensitivity factor covering 99.7% of the normal distribution. Theoretically, any score exceeding this threshold is statistically highly unlikely to be a benign fluctuation (with a probability of less than 0.3%) (Pukelsheim, 1994).

**Uncertainty-modulated final threshold.** Subsequently, we relax the strict statistical bound when the ego vehicle itself is uncertain. The final adaptive threshold $\tau_t$ is modulated by the uncertainty term $U_t$:

$$\tau_t = T_{stat} + \beta \cdot U_t \cdot (T_{max} - T_{stat}), \qquad (11)$$

where $\beta$ controls the impact of uncertainty, and $T_{max}$ is the upper safety bound. This formulation ensures that in high-uncertainty scenarios ($U_t \to 1$), the threshold expands towards $T_{max}$ to prevent false positives (misidentifying benign CAVs as attackers), while in stable scenarios, it converges to the strict statistical bound $T_{stat}$.

*Table 1.* Performance comparison of different defense methods under MOR and TOR attacks across multiple CP models.

| Attack | Method | No Defense | | | ROBOSAC | | | LUCIA | | | CP-Guard | | | Cerberus (Ours) | | |
|---|---|---|---|---|---|---|---|---|---|---|---|---|---|---|---|---|
| | Model | ORR↓/ASR↓ | AP@0.5 | AP@0.7 | ORR↓/ASR↓ | AP@0.5 | AP@0.7 | ORR↓/ASR↓ | AP@0.5 | AP@0.7 | ORR↓/ASR↓ | AP@0.5 | AP@0.7 | ORR↓/ASR↓ | AP@0.5 | AP@0.7 |
| MOR | AttFusion | 100%/99.95% | 0.00 | 0.00 | 10.52%/0.20% | 0.71 | 0.55 | **7.52%/0.15%** | 0.84 | 0.72 | 7.82%/0.20% | 0.84 | 0.72 | 7.54%/**0.15%** | 0.84 | 0.72 |
| | CoAlign | 99.99%/99.76% | 0.00 | 0.00 | 8.03%/0.05% | 0.72 | 0.59 | **5.73%/0.05%** | 0.85 | 0.76 | 5.83%/0.05% | 0.85 | 0.76 | **5.73%/0.05%** | 0.85 | 0.76 |
| | Where2comm | 94.26%/55.24% | 0.00 | 0.00 | 5.65%/0.10% | 0.75 | 0.51 | **4.52%**/0.10% | 0.85 | 0.65 | 4.65%/0.10% | 0.85 | 0.66 | 4.53%/**0.05%** | 0.85 | 0.66 |
| | V2VAM | 100%/100% | 0.00 | 0.00 | 12.97%/0.29% | 0.77 | 0.63 | 19.09%/0.39% | 0.82 | 0.76 | 8.27%/0.30% | 0.87 | 0.79 | **7.87%/0.19%** | 0.88 | 0.79 |
| TOR (In) | AttFusion | −/99.61% | 0.89 | 0.79 | −/96.34% | 0.89 | 0.78 | −/6.05% | 0.84 | 0.72 | −/97.42% | 0.89 | 0.79 | −/**4.19%** | 0.84 | 0.72 |
| | CoAlign | −/99.12% | 0.89 | 0.80 | −/96.83% | 0.88 | 0.80 | −/5.07% | 0.85 | 0.76 | −/96.64% | 0.89 | 0.81 | −/**2.97%** | 0.85 | 0.76 |
| | Where2comm | −/78.21% | 0.20 | 0.03 | −/18.33% | 0.71 | 0.46 | −/5.31% | 0.85 | 0.66 | −/19.89% | 0.72 | 0.49 | −/**3.31%** | 0.85 | 0.66 |
| | V2VAM | −/97.37% | 0.81 | 0.66 | −/92.89% | 0.81 | 0.66 | −/14.14% | 0.82 | 0.76 | −/93.95% | 0.81 | 0.66 | −/**4.48%** | 0.88 | 0.79 |
| TOR (Out) | AttFusion | −/98.79% | 0.90 | 0.79 | −/96.78% | 0.89 | 0.78 | −/47.25% | 0.84 | 0.72 | −/97.32% | 0.89 | 0.79 | −/**32.62%** | 0.84 | 0.72 |
| | CoAlign | −/98.39% | 0.88 | 0.80 | −/98.49% | 0.88 | 0.79 | −/40.03% | 0.85 | 0.76 | −/96.64% | 0.88 | 0.80 | −/**26.81%** | 0.85 | 0.76 |
| | Where2comm | −/77.76% | 0.20 | 0.03 | −/58.81% | 0.70 | 0.45 | −/28.08% | 0.86 | 0.65 | −/55.29% | 0.70 | 0.47 | −/**15.26%** | 0.85 | 0.66 |
| | V2VAM | −/97.22% | 0.82 | 0.66 | −/97.56% | 0.82 | 0.66 | −/62.32% | 0.82 | 0.76 | −/94.83% | 0.82 | 0.66 | −/**38.22%** | 0.88 | 0.79 |
| TOR (Random) | AttFusion | −/98.09% | 0.90 | 0.79 | −/95.90% | 0.89 | 0.78 | −/9.02% | 0.84 | 0.72 | −/95.32% | 0.89 | 0.79 | −/**8.92%** | 0.84 | 0.72 |
| | CoAlign | −/97.22% | 0.88 | 0.80 | −/95.07% | 0.88 | 0.79 | −/**6.73%** | 0.85 | 0.76 | −/95.56% | 0.88 | 0.79 | −/**6.73%** | 0.85 | 0.76 |
| | Where2comm | −/58.56% | 0.21 | 0.03 | −/16.86% | 0.71 | 0.46 | −/5.52% | 0.85 | 0.66 | −/22.38% | 0.72 | 0.49 | −/**5.16%** | 0.85 | 0.66 |
| | V2VAM | −/94.93% | 0.81 | 0.66 | −/89.62% | 0.82 | 0.66 | −/20.33% | 0.82 | 0.75 | −/91.91% | 0.82 | 0.66 | −/**9.56%** | 0.88 | 0.79 |

**Defense execution.** Finally, we make the detection decision based on the adaptive threshold $\tau_t$ at frame $t$. Specifically, for each collaborator group, if its anomaly score satisfies $J_k < \tau_t$, the group is classified as benign. Otherwise, the group is considered to contain an attacker, and we further partition it recursively and continue the feature-level consistency verification until the malicious vehicle is located. All identified benign vehicles are ultimately accepted by the ego vehicle for CP, and they also serve as the basis for threshold computation in the next frame.

# 5. Experiments

## 5.1. Experimental Setups

Following (Wang et al., 2025), we evaluate Cerberus on the OPV2V benchmark (Xu et al., 2022). We compare our method against state-of-the-art attacks (MOR and TOR (Wang et al., 2025)) and classical adversarial attacks (BIM, PGD, and C&W (Hu et al., 2025b)). For TOR, we assess three variants where targets are positioned inside (In), outside (Out), or randomly relative to the victim's field of view (Random). To assess generality, we employ four fusion models: AttFusion (Xu et al., 2022), Where2comm (Hu et al., 2022), CoAlign (Lu et al., 2023), and V2VAM (Li et al., 2023a). Comparisons are made against ROBOSAC (Li et al., 2023b), LUCIA (Wang et al., 2025), and CP-Guard (Hu et al., 2025b) using Attack Success Rate (ASR), Object Removal Rate (ORR), and AP@0.5/0.7. Detailed experimental settings are provided in the appendix B.

## 5.2. Defense Effectiveness

**Defense against state-of-the-art attacks.** To effectively evaluate the defense capability of Cerberus against state-of-the-art attacks, we comparatively tested its performance under MOR and TOR attacks. As presented in Table 1, Cerberus demonstrates exceptional resilience against

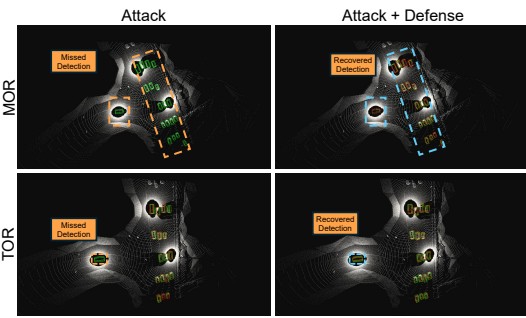

*Figure 3.* Visualization of Cerberus defense performance, where green boxes indicate ground truth and red boxes denote detections. Vehicles removed by the attack are effectively recovered.

MOR, restoring V2VAM's AP@0.5 to 0.88 with only 7.87% ORR, surpassing baselines like CP-Guard and LUCIA. The robustness of our method becomes even more pronounced under TOR attacks. Conversely, ROBOSAC and CP-Guard are rendered ineffective against TOR attacks due to their inability to detect such stealthy perturbations. In contrast, Cerberus consistently achieves low ASR across all experimental settings. These results demonstrate that, compared to methods relying on output-level verification, Cerberus effectively defends against stealthy adversarial attacks by enforcing consistency checks on deep feature maps. The visualization in Figure 3 further validates this conclusion.

**Defense against classical attacks.** Table 2 shows defense performance under BIM, PGD, and C&W attacks. As observed, the no defense baseline fails catastrophically across all models with zero AP, highlighting CP systems' vulnerability to feature perturbations. In contrast, Cerberus consistently demonstrates superior robustness, effectively restoring detection capabilities to a high level. While RO-BOSAC provides basic protection, it falls significantly behind in detection accuracy, achieving only 0.77/0.63 on V2VAM compared to our 0.88/0.79. LUCIA and CP-Guard

*Table 2.* Comparison of the defense performance of different methods under BIM, PGD, and C&W attacks.

| Attack | Method | No Defense | | ROBOSAC | | LUCIA | | CP-Guard | | Cerberus (Ours) | |
|---|---|---|---|---|---|---|---|---|---|---|---|
| | Model | AP@0.5 | AP@0.7 | AP@0.5 | AP@0.7 | AP@0.5 | AP@0.7 | AP@0.5 | AP@0.7 | AP@0.5 | AP@0.7 |
| BIM | AttFusion | 0 | 0 | 0.71 | 0.55 | **0.84** | **0.72** | 0.83 | 0.71 | **0.84** | **0.72** |
| | CoAlign | 0 | 0 | 0.72 | 0.59 | **0.85** | **0.76** | 0.85 | 0.76 | 0.85 | 0.76 |
| | Where2comm | 0 | 0 | 0.75 | 0.51 | **0.85** | 0.65 | 0.85 | 0.65 | 0.85 | **0.66** |
| | V2VAM | 0 | 0 | 0.77 | 0.63 | 0.82 | 0.76 | 0.87 | **0.79** | 0.88 | 0.79 |
| PGD | AttFusion | 0 | 0 | 0.71 | 0.55 | **0.84** | **0.72** | 0.83 | 0.71 | 0.84 | 0.72 |
| | CoAlign | 0 | 0 | 0.72 | 0.59 | **0.85** | **0.76** | 0.85 | 0.76 | 0.85 | 0.76 |
| | Where2comm | 0 | 0 | 0.75 | 0.51 | **0.85** | **0.65** | 0.85 | 0.65 | 0.85 | 0.65 |
| | V2VAM | 0 | 0 | 0.77 | 0.64 | 0.82 | 0.76 | 0.87 | **0.79** | 0.88 | 0.79 |
| C&W | AttFusion | 0 | 0 | 0.71 | 0.55 | **0.84** | **0.72** | 0.83 | 0.71 | 0.84 | 0.72 |
| | CoAlign | 0 | 0 | 0.72 | 0.59 | **0.85** | **0.76** | 0.85 | 0.76 | 0.85 | 0.76 |
| | Where2comm | 0 | 0 | 0.75 | 0.51 | **0.85** | 0.65 | 0.85 | 0.65 | 0.85 | **0.66** |
| | V2VAM | 0 | 0 | 0.77 | 0.64 | 0.82 | 0.76 | 0.87 | **0.79** | 0.88 | 0.79 |

*Table 3.* Ablation study of Cerberus components under MOR and TOR attacks.

| Attack | MOR | | | TOR | | |
|---|---|---|---|---|---|---|
| Method Variant | ORR↓/ASR↓ | AP@0.5 | AP@0.7 | ASR↓ | AP@0.5 | AP@0.7 |
| $D_{topo}$ | 8.37%/0.20% | 0.80 | 0.67 | 9.46% | 0.80 | 0.67 |
| $D_{sem}$ | 7.94%/0.15% | 0.84 | 0.72 | 9.07% | 0.84 | 0.72 |
| $D_{mag}$ | 7.69%/0.15% | 0.84 | 0.72 | 9.12% | 0.84 | 0.72 |
| Cerberus(Full) | **7.54%/0.15%** | 0.84 | 0.72 | **8.92%** | 0.84 | 0.72 |

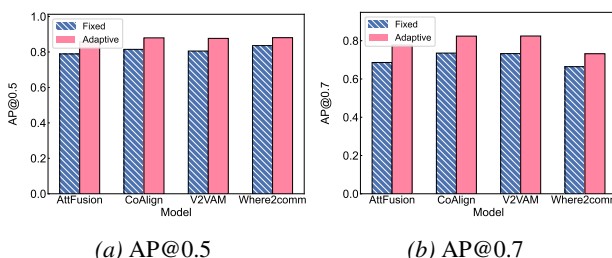

*(a)* AP@0.5      *(b)* AP@0.7

*Figure 4.* Effectiveness of the uncertainty-aware adaptive decision module. Comparison of AP using fixed vs. adaptive thresholds. Results on (a) AP@0.5 and (b) AP@0.7 demonstrate that our module consistently outperforms the fixed baseline across all models.

demonstrate strong competitiveness, but Cerberus consistently matches or surpasses them across all metrics. Notably, under the Where2comm model, Cerberus achieves an AP@0.7 of 0.66, surpassing CP-Guard and LUCIA. This indicates that our multi-dimensional consistency check is more effective at preserving high-quality fusion features while filtering out subtle adversarial noise.

### 5.3. Ablation Study

**Impact of consistency verification.** As shown in Table 3, we evaluate the contribution of each consistency metric to the defense performance. It can be observed that the variant relying solely on topological consistency ($D_{topo}$) exhibits the weakest defensive capability, with an ORR of 8.37% under MOR attacks. This indicates that structural verification alone is insufficient to fully filter out complex adversarial perturbations. Incorporating semantic ($D_{sem}$) or energy ($D_{mag}$) dimensions enhances robustness, reducing the ORR to 7.94% and 7.69% respectively. Notably, the full Cerberus framework achieves the best overall performance, further lowering the ORR to 7.54% while maintaining the highest detection precision. These results indicate that each component of Cerberus contributes positively to defending against diverse attacks, demonstrating the effectiveness of the individual components.

**Impact of uncertainty-aware adaptive decision.** To validate performance in realistic scenarios, we introduced a

mixed setting involving both active attackers and benign agents subject to environmental noise (simulated on 50% of participants). As illustrated in Figure 4, the static threshold strategy struggles to adapt to such complex environments, leading to a significant degradation in the performance of the fixed-threshold Cerberus variant. In contrast, Cerberus demonstrates superior resilience, significantly outperforming fixed baselines. This is because our uncertainty-aware adaptive decision module dynamically calibrates the rejection boundary based on estimated uncertainty, effectively distinguishing between benign environmental degradation and malicious perturbations even when half the network is compromised by noise.

### 5.4. Efficiency and Robustness Analysis

**Defense against adaptive attacker.** To effectively evaluate the defense capability of Cerberus against adaptive attacks, we consider an attacker who has full knowledge of our defense logic and parameter settings. This attacker is dedicated to minimizing the joint anomaly score of Cerberus while achieving the attack objectives. To fabricate adversarial perturbations, the attacker employs a PGD-based op-

*Table 4.* Defense performance of Cerberus against adaptive attacks. Cerberus demonstrates strong defensive capabilities, significantly reducing ASR and ORR of MOR and TOR attacks while maintaining high AP.

| Attack | | MOR | | TOR | | |
|---|---|---|---|---|---|---|
| Model | ORR↓/ASR↓ | AP@0.5 | AP@0.7 | ASR↓ | AP@0.5 | AP@0.7 |
| AttFusion | 7.53%/0.15% | 0.84 | 0.72 | 9.02% | 0.84 | 0.72 |
| CoAlign | 5.72%/0.05% | 0.85 | 0.76 | 6.72% | 0.85 | 0.76 |
| Where2comm | 4.55%/0.09% | 0.85 | 0.66 | 5.26% | 0.85 | 0.66 |
| V2VAM | 8.36%/0.24% | 0.87 | 0.79 | 9.84% | 0.87 | 0.79 |

*Table 5.* Robustness against adaptive attacks under different settings.

| Surrogate | Iter. | Starts | MOR | | TOR | |
|---|---|---|---|---|---|---|
| | | | ASR/ORR ↓ | AP@0.5/0.7 ↑ | ASR ↓ | AP@0.5/0.7 ↑ |
| Joint | 10 | 1 | 0.15%/7.53% | 0.84/0.72 | 9.02% | 0.84/0.72 |
| Joint | 20 | 1 | 0.20%/8.04% | 0.84/0.72 | 9.17% | 0.84/0.72 |
| Joint | 20 | 5 | 0.20%/8.05% | 0.84/0.72 | 9.18% | 0.84/0.72 |
| Grouping-aware | 20 | 1 | 0.20%/8.06% | 0.84/0.72 | 9.17% | 0.84/0.72 |
| Grouping-aware | 20 | 5 | 0.20%/8.05% | 0.83/0.71 | 9.18% | 0.83/0.71 |

timization with a learning rate of 0.1 over 10 iterations to launch attacks adaptively. Table 4 presents the defense performance of Cerberus under such adaptive attacks. Experimental results demonstrate that Cerberus maintains robust defensive capabilities even against such sophisticated adversaries. Across various collaborative frameworks (AttFusion, CoAlign, Where2comm, and V2VAM), our method effectively suppresses the ASR and ORR to low levels (e.g., < 10% for TOR) while preserving high detection precision. Notably, with V2VAM, Cerberus achieves an impressive AP@0.5 of 0.87, indicating that our multi-head verification mechanism successfully constrains the feasible space for adaptive perturbations.

**Robustness against varying adaptive attack configurations.** To further investigate whether stronger or differently configured adaptive attacks can bypass our defense, we vary the surrogate type, number of optimization iterations, and random restarts, as shown in Table 5. Regardless of these configurations, our defense maintains stable performance throughout. With the joint surrogate, increasing iterations from 10 to 20 or adding 5 random restarts causes no observable degradation, with ASR/ORR remaining around 0.20%/8.05% and AP@0.5/0.7 fixed at 0.84/0.72. Replacing the surrogate with a grouping-aware model, which is tailored to exploit the internal structure of our defense, similarly yields no meaningful gain for the attacker. Even under the most aggressive setting with a grouping-aware surrogate, 20 iterations, and 5 random restarts, the AP decreases by merely 0.01, confirming that our defense does not become more vulnerable as attack strength increases.

**Efficiency-robustness trade-off.** To comprehensively evaluate the defense performance of Cerberus, we benchmarked it against state-of-the-art defense methods under stealthy attacks that perturb local regions of the feature

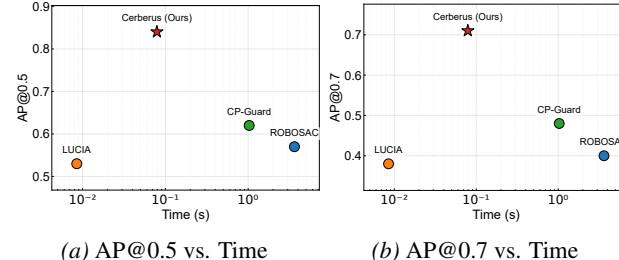

*(a)* AP@0.5 vs. Time   *(b)* AP@0.7 vs. Time

*Figure 5.* Accuracy vs. Latency trade-off. Comparison of detection performance (AP) against defense computation time (log scale) for (a) AP@0.5 and (b) AP@0.7.

*Table 6.* Latency of each component in our Cerberus (ms).

| Component | $N$=5 | $N$=10 | $N$=15 | $N$=20 | $N$=30 | $N$=40 |
|---|---|---|---|---|---|---|
| Grouping | 0.86 | 3.08 | 6.64 | 11.09 | 20.74 | 30.97 |
| Topological | 1.66 | 2.36 | 2.55 | 2.62 | 2.58 | 2.65 |
| Semantic | 19.48 | 28.15 | 30.76 | 32.00 | 33.14 | 34.29 |
| Energy | 0.88 | 1.22 | 1.37 | 1.43 | 1.32 | 1.34 |

maps. Figure 5 visualizes the trade-off between detection performance (AP) and computational latency (in seconds, log scale). As observed, LUCIA achieves the lowest latency ($\sim 10^{-2}$s). This is because it computes simple statistical metrics (e.g., pooled $L_1$ distance) on compressed features. However, this efficiency comes at the cost of a catastrophic drop in defense performance. Since LUCIA performs verification on spatially compressed features (i.e., losing local details), it fails to detect subtle, local perturbations in the feature space, leading to system vulnerabilities. Conversely, traditional methods like CP-Guard and ROBOSAC suffer from prohibitive computational overheads, making them unsuitable for real-time deployment. In contrast, Cerberus strikes an optimal trade-off. Although incurring a marginal latency increase compared to LUCIA ($\sim 10^{-1}$s), our method achieves superior detection accuracy.

**Component-wise latency analysis.** Table 6 further breaks down the latency contribution of each component in our defense pipeline. The semantic module accounts for the majority of the overall latency, stabilizing at approximately 30–34 ms beyond $N$=15, indicating that its complexity is largely independent of the number of vehicles. The topological module remains nearly constant across all CAV counts, consistently below 2.65 ms, demonstrating strong scalability. The grouping module exhibits mild growth with the number of CAVs, which is expected given its graph-based construction process. The energy module introduces negligible overhead throughout. Overall, the latency of each component remains well-controlled, confirming the efficiency of our design.

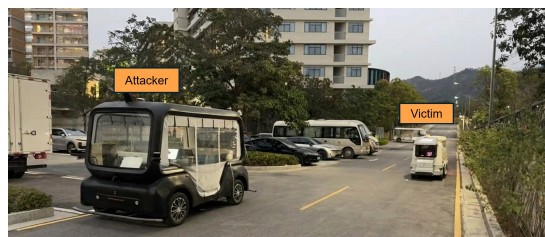

*Figure 6.* Real-world collaborative perception testbed.

## 5.5. Real-World Evaluation

**System setup.** As illustrated in Figure 6, we construct a physical CP testbed consisting of two autonomous vehicles. The victim (ego) vehicle is equipped with a RoboSense Helios-32 main LiDAR, three Bpearl-32 blind-spot LiDARs, and a high-precision GNSS/RTK system. The attacker vehicle operates with a RoboSense Helios-16 main LiDAR, supplemented by two Helios-32P and one E1 solid-state LiDAR, localized via an integrated inertial navigation system.

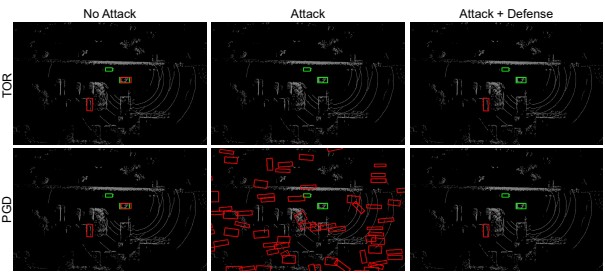

*Figure 7.* Visualization of the defense performance of `Cerberus` in real-world scenarios under TOR and PGD attacks. After deploying `Cerberus`, the adverse effects caused by the attacks are effectively mitigated.

**Visualization of real-world defense performance.** To effectively evaluate the real-world effectiveness of our defense framework, we assess its performance under TOR and PGD attacks. As shown in Figure 7, these attacks significantly degrade the perception outputs. However, `Cerberus` effectively identifies and purges these anomalies in the latent space. The resulting output is a clean detection result that preserves only real vehicles, demonstrating the robustness of our method against adversarial perturbations in the physical world.

## 6. Conclusion

In this paper, we propose `Cerberus`, which is a novel defense framework designed to counter adversarial attacks in CP systems by leveraging multi-dimensional consistency within the feature space. Specifically, `Cerberus` employs isomorphic view fusion to address geometric misalignment and consistency verification to identify feature anomalies across multiple dimensions. Furthermore, it integrates an uncertainty-aware adaptive decision module to dynamically calibrate detection boundaries for enhanced robustness. Extensive experiments on both datasets and in the physical world demonstrate the effectiveness of `Cerberus`.

## Acknowledgements

The work is supported in part by a project from Hong Kong Research Grant Council under Research Impact Fund (RIF) R1012-21, and the National Natural Science Foundation of China under Grant 62441237, Grant U24A20336, Grant 62272348, and Grant U22B2022.

## Impact Statement

This paper addresses critical security vulnerabilities in CP for autonomous driving. Our proposed method, `Cerberus`, significantly improves system resilience against adversarial attacks that attempt to hide objects from connected vehicles, directly contributing to the safety and reliability of intelligent transportation systems. While the study of adversarial attacks carries inherent dual-use risks, understanding these threats is essential for developing effective defenses. All experiments were performed in a controlled simulation environment using offline datasets, ensuring no physical risk to human safety or public infrastructure.

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

# A. Detailed Implementation of Our `Cerberus`

To facilitate a deeper understanding and ensure the reproducibility of our proposed `Cerberus`, this section provides the formal procedural description. The algorithm follows the architecture illustrated in Algorithm 1, specifically focusing on the computational flow of the three core components: (1) Isomorphic View Fusion, (2) Feature-Level Consistency Verification, and (3) Uncertainty-Aware Adaptive Decision.

---

**Algorithm 1** Main Workflow of `Cerberus`

---

1: **Input:** Ego feature $\mathbf{F}_{ego}$, feature maps from CAVs $\mathbf{F}_0, \mathbf{F}_1, ..., \mathbf{F}_N$, initial threshold $\tau_0$, Candidate groups $\mathbb{G}$, History Window $\mathcal{W}$, and anomaly score window size $\kappa$
2: **Output:** Benign fusion candidates $\mathbb{G}_{benign}$
3: // Phase 1: Isomorphic View Fusion
4: Obtaining two viewpoint-isomorphic vehicle groups $\mathbb{G}_{1,2}$
5: $\mathbf{F}_{\mathbb{G}_{1,2}} = \mathcal{F}_{fusion}\left(\{\mathbf{F}_j\}_{j \in \mathbb{G}_{1,2}} \mid \Theta\right)$
6: // Phase 2: Feature-Level Consistency Verification
7: Initialize $\mathbb{G}_{benign} \leftarrow \emptyset$
8: **for** each group $\mathbb{G}_k$ **in** $\mathbb{G}$ **do**
9:    $S(\mathbf{F}_{\mathbb{G}_k}, \mathbf{F}_{ego}) = \frac{(2\mu_k\mu_{ego}+C_1)(2\sigma_{k,ego}+C_2)}{(\mu_k^2+\mu_{ego}^2+C_1)(\sigma_k^2+\sigma_{ego}^2+C_2)}$
10:    $D_{topo} = 1 - S(F_{\mathbb{G}_k}, F_{ego})$
11:    $D_{sem} = 1 - \frac{\mathbf{F}_{\mathbb{G}_k} \cdot \mathbf{F}_{ego}}{\|\mathbf{F}_{\mathbb{G}_k}\|_2 \|\mathbf{F}_{ego}\|_2}$
12:    $D_{mag} = \exp\left(\frac{\|\mathbf{F}_{ego}-\mathbf{F}_{\mathbb{G}_k}\|_1}{\sigma}\right)$
13:    Compute score $J_k = \lambda_1 D_{topo} + \lambda_2 D_{sem} + \lambda_3 D_{mag}$
14:    **if** $J_k > \tau_t$ **then**
15:      Continue the recursive consistency verification
16:    **else**
17:      Add $\mathbb{G}_k$ to $\mathbb{G}_{benign}$
18:      Update window $\mathcal{W} \leftarrow \mathcal{W} \cup \{J_k\}$
19:    **end if**
20: **end for**
21: **if** $|\mathcal{W}| > N_{window}$ **then**
22:    Remove oldest entry from $\mathcal{W}$
23: **end if**
24: // Phase 3: Uncertainty-Aware Adaptive Decision
25: Compute uncertainty $U_t = \frac{-1}{\log(C)} \sum_{c=1}^{C} p(f_{ego}^c) \log p(f_{ego}^c)$
26: **if** $|\mathcal{W}| \geq \kappa$ **then**
27:    $\mu_t, \sigma_t \leftarrow \text{Mean}(\mathcal{W}), \text{Std}(\mathcal{W})$
28:    $T_{stat} \leftarrow \mu_t + \gamma \cdot \sigma_t$
29: **else**
30:    $T_{stat} \leftarrow T_0$
31: **end if**
32: Calculate final threshold $\tau_t$:
33: $\tau_t \leftarrow T_{stat} + \beta \cdot U_t \cdot (T_{max} - T_{stat})$
34: **Return** $\mathbb{G}_{benign}$

---

**Theorem A.1** (Angular Equilibrium of Interleaved Allocation). *Let the azimuth angles of $N$ neighboring vehicles be arbitrarily distributed over $[-\pi, \pi]$, and let $N_{sorted}$ be the sequence sorted by azimuth. Define the maximum adjacent angular difference as:*

$$\Delta_{\max} = \max_{1 \leq i < N} (\theta_{i+1} - \theta_i). \tag{12}$$

*After interleaving the vehicles into two groups $G_1$ and $G_2$, the angular difference between correspondingly positioned elements satisfies:*

$$|\theta_{G_1}^{(k)} - \theta_{G_2}^{(k)}| \leq \Delta_{\max}. \tag{13}$$

*Specifically, when the $N$ vehicles are uniformly distributed over $[-\pi, \pi]$, the difference satisfies:*

$$|\theta_{G_1}^{(k)} - \theta_{G_2}^{(k)}| \leq \frac{2\pi}{N} \xrightarrow{N \to \infty} 0. \tag{14}$$

*Proof.* After sorting all $N$ vehicles by azimuth angle, we obtain the ordered sequence $\theta_1 \leq \theta_2 \leq \cdots \leq \theta_N$. The interleaved allocation assigns odd-indexed vehicles to $G_1$ and even-indexed vehicles to $G_2$, i.e.,

$$G_1 = \{\theta_1, \theta_3, \ldots\}, \quad G_2 = \{\theta_2, \theta_4, \ldots\}. \tag{15}$$

The $k$-th correspondingly positioned pair is $(\theta_{2k-1}, \theta_{2k})$, which are adjacent in the sorted sequence. Therefore:

$$|\theta_{G_1}^{(k)} - \theta_{G_2}^{(k)}| = \theta_{2k} - \theta_{2k-1} \leq \max_{1 \leq i < N}(\theta_{i+1} - \theta_i) = \Delta_{\max}. \tag{16}$$

When the $N$ vehicles are uniformly distributed over $[-\pi, \pi]$, the angular spacing between adjacent vehicles is $\frac{2\pi}{N}$, so $\Delta_{\max} = \frac{2\pi}{N}$, and thus:

$$|\theta_{G_1}^{(k)} - \theta_{G_2}^{(k)}| \leq \frac{2\pi}{N} \xrightarrow{N \to \infty} 0. \tag{17}$$

This completes the proof. $\square$

**Theorem A.2** (False Positive Rate Upper Bound of the Adaptive Threshold). *Assume the anomaly scores of benign vehicles within a statistical window follow a Gaussian distribution $J_k \sim \mathcal{N}(\mu_t, \sigma_t^2)$, and let $T_{\text{stat}} = \mu_t + 3\sigma_t$ be the static threshold. Given the adaptive threshold $\tau_t = T_{\text{stat}} + \beta U_t(T_{\max} - T_{\text{stat}})$, where $U_t \geq 0$, $\beta > 0$, and $T_{\max} > T_{\text{stat}}$, the false positive rate satisfies:*

$$P(\text{FP} \mid U_t) = 1 - \Phi\left(3 + \frac{\beta U_t(T_{\max} - T_{\text{stat}})}{\sigma_t}\right) \leq 1 - \Phi(3), \tag{18}$$

*and monotonically decreases as $U_t$ increases.*

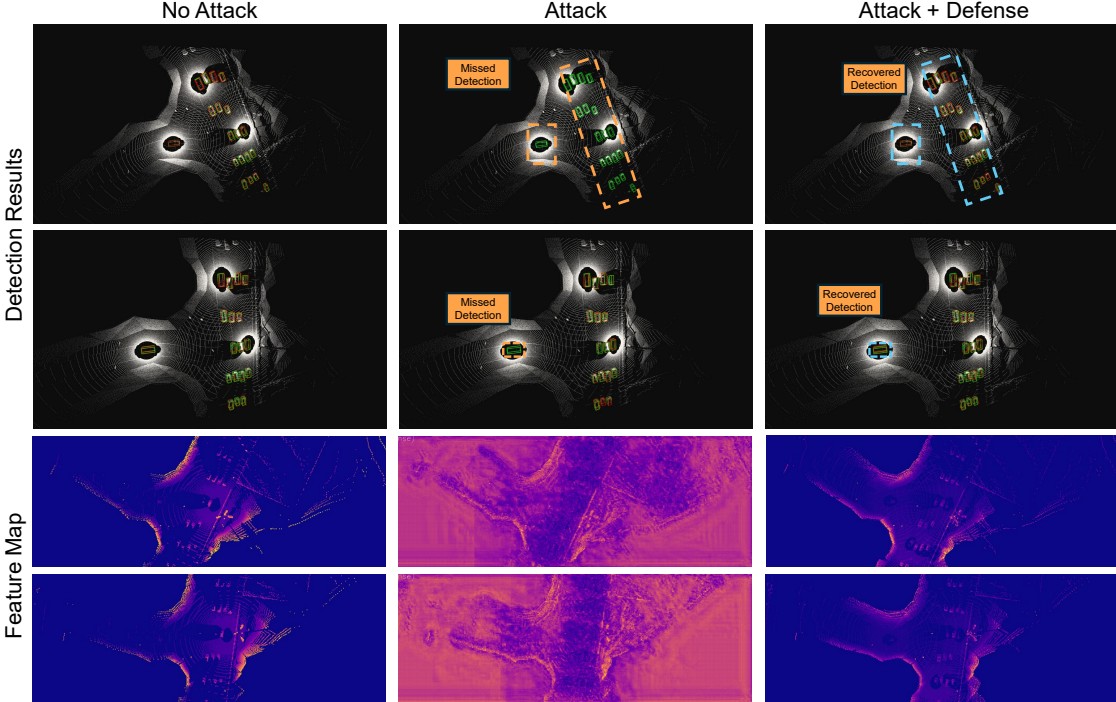

*Figure 8.* Visualization of `Cerberus` defense performance. Green bounding boxes denote ground truth annotations, while red bounding boxes indicate objects detected by the model. The results demonstrate that `Cerberus` effectively restores accurate detection results and preserves high-fidelity feature representations.

*Proof.* Since $\tau_t = T_{\text{stat}} + \beta U_t(T_{\text{max}} - T_{\text{stat}})$ and $J_k \sim \mathcal{N}(\mu_t, \sigma_t^2)$, the false positive rate is:

$$
\begin{aligned}
P(\text{FP} \mid U_t) &= P(J_k > \tau_t) \\
&= 1 - \Phi\left(\frac{\tau_t - \mu_t}{\sigma_t}\right) \\
&= 1 - \Phi\left(3 + \frac{\beta U_t(T_{\text{max}} - T_{\text{stat}})}{\sigma_t}\right). \quad (19)
\end{aligned}
$$

Since $\Phi(\cdot)$ is monotonically increasing and $\beta U_t(T_{\text{max}} - T_{\text{stat}}) \geq 0$, the argument is no less than 3, yielding:

$$
P(\text{FP} \mid U_t) \leq 1 - \Phi(3). \quad (20)
$$

Moreover, as $U_t$ increases, the argument of $\Phi$ strictly increases, so $P(\text{FP} \mid U_t)$ monotonically decreases. □

## B. More Details of the Experimental Results

**Dataset and models.** Similar to (Wang et al., 2025), we evaluated our proposed defense method using the OPV2V dataset (Xu et al., 2022). OPV2V is a large-scale benchmark for V2V perception, collected across 70 diverse scenes from 8 towns in the CARLA digital twin simulator (Dosovitskiy et al., 2017). The dataset comprises 11,464 frames of LiDAR point cloud data captured from 2-5 CAVs, along with 232,913 annotated 3D vehicle bounding boxes. In our

experiments, we designate CAV 0 as the victim and CAV 1 as the attacker, aligning with the configuration in (Wang et al., 2025). Regarding attack configurations, we employ a diverse set of methods including two state-of-the-art attacks, TOR and MOR (Wang et al., 2025), as well as three classical adversarial benchmarks, namely BIM, PGD, and C&W. The TOR attack includes three variants, denoted as TOR(in), TOR(out), and TOR(random), which correspond to scenarios where the target object is within the victim vehicle's field of view, outside the victim's field of view, and randomly positioned, respectively. We further evaluate these attacks across four representative intermediate fusion models, namely AttFusion (Xu et al., 2022), Where2comm (Hu et al., 2022), CoAlign (Lu et al., 2023), and V2VAM (Li et al., 2023a), to ensure the generalizability of our findings.

**Baselines.** To evaluate the performance of `Cerberus`, we compare it with three representative defense baselines: ROBOSAC (Li et al., 2023b), LUCIA (Wang et al., 2025), and CP-Guard (Hu et al., 2025b). ROBOSAC is a consensus-based defense method that follows a hypothesis-and-verification paradigm. By repeatedly sampling CAVs, it identifies vehicles that exhibit consistent perception outcomes, thereby filtering out malicious vehicles. LUCIA (Wang et al., 2025) is a defense method based on a trustworthiness-aware attention mechanism. It mitigates adversarial attacks by assigning different weights to messages

*Table 7.* Performance impact of individual components under various attacks. The results show that the full version of our framework achieves the best defense performance, demonstrating that all three components are essential.

| Model | Attack | MOR | | | TOR(In) | | | TOR(Out) | | | TOR(Random) | | |
|---|---|---|---|---|---|---|---|---|---|---|---|---|---|
| | Method Variant | ORR↓/ASR↓ | AP@0.5 | AP@0.7 | ASR↓ | AP@0.5 | AP@0.7 | ASR↓ | AP@0.5 | AP@0.7 | ASR↓ | AP@0.5 | AP@0.7 |
| AttFusion | $D_{topo}$ | 8.37%/0.20% | 0.8 | 0.67 | 4.63% | 0.8 | 0.67 | 33.44% | 0.8 | 0.67 | 9.46% | 0.8 | 0.67 |
| | $D_{sem}$ | 7.94%/0.15% | 0.84 | 0.72 | 4.19% | 0.84 | 0.72 | 32.42% | 0.84 | 0.72 | 9.07% | 0.84 | 0.72 |
| | $D_{mag}$ | 7.69%/0.15% | 0.84 | 0.72 | 4.24% | 0.84 | 0.72 | 33.16% | 0.84 | 0.72 | 9.12% | 0.84 | 0.72 |
| | Cerberus(Full) | **7.54%/0.15%** | 0.84 | 0.72 | **4.19%** | 0.84 | 0.72 | **32.62%** | 0.84 | 0.72 | **8.92%** | 0.84 | 0.72 |
| CoAlign | $D_{topo}$ | 85.57%/84.98% | 0.16 | 0.14 | 83.08% | 0.87 | 0.78 | 85.76% | 0.87 | 0.77 | 83.37% | 0.87 | 0.78 |
| | $D_{sem}$ | 5.68%/0.05% | 0.85 | 0.76 | 2.97% | 0.85 | 0.76 | 27.16% | 0.85 | 0.76 | 6.78% | 0.85 | 0.76 |
| | $D_{mag}$ | 5.78%/0.05% | 0.85 | 0.76 | 2.97% | 0.85 | 0.76 | 27.25% | 0.85 | 0.76 | 6.78% | 0.85 | 0.76 |
| | Cerberus(Full) | **5.73%/0.05%** | 0.85 | 0.76 | 2.97% | 0.85 | 0.76 | **26.81%** | 0.85 | 0.76 | **6.73%** | 0.85 | 0.76 |
| Where2comm | $D_{topo}$ | 4.71%/0.10% | 0.83 | 0.62 | **2.93%** | 0.83 | 0.62 | **14.82%** | 0.83 | 0.62 | **5.07%** | 0.83 | 0.62 |
| | $D_{sem}$ | 4.56%/0.10% | 0.85 | 0.65 | 3.32% | 0.85 | 0.66 | 15.46% | 0.85 | 0.65 | 5.17% | 0.85 | 0.65 |
| | $D_{mag}$ | 4.62%/0.10% | 0.85 | 0.65 | 3.32% | 0.85 | 0.65 | 15.26% | 0.85 | 0.65 | 5.22% | 0.85 | 0.65 |
| | Cerberus(Full) | **4.53%/0.05%** | 0.85 | 0.66 | 3.31% | 0.85 | 0.66 | 15.26% | 0.85 | 0.66 | 5.16% | 0.85 | 0.66 |
| V2VAM | $D_{topo}$ | 86.03%/85.08% | 0.17 | 0.14 | 85.23% | 0.81 | 0.65 | 89.13% | 0.82 | 0.66 | 84.98% | 0.80 | 0.65 |
| | $D_{sem}$ | 7.88%/0.20% | 0.88 | 0.79 | 4.49% | 0.88 | 0.79 | 38.98% | 0.88 | 0.79 | 9.66% | 0.88 | 0.79 |
| | $D_{mag}$ | 8.14%/0.24% | 0.88 | 0.79 | 4.58% | 0.88 | 0.79 | 38.52% | 0.88 | 0.79 | 9.61% | 0.88 | 0.79 |
| | Cerberus(Full) | **7.87%/0.19%** | 0.88 | 0.79 | **4.48%** | 0.88 | 0.79 | **38.22%** | 0.88 | 0.79 | **9.56%** | 0.88 | 0.79 |

from various CAVs. CP-Guard (Hu et al., 2025b) is also a consensus-based defense method that identifies benign and malicious vehicles by recursively partitioning and sampling CAVs.

**Evaluation metrics.** To comprehensively evaluate the effectiveness of our proposed Cerberus, we employed four key metrics: Attack Success Rate (ASR), Object Removal Rate (ORR), AP@0.5, and AP@0.7. The ASR quantifies the fraction of frames in the dataset where the adversarial attack successfully removes the targeted object(s) from the perception model's detections. It reflects the effectiveness of the attack in causing the perception system to miss the targeted objects. Its formal definition is:

$$ASR = \frac{N_s}{N},$$  (21)

where $N_s$ denotes the number of frames where the attack succeeds, and $N$ is the total number of frames. For Mass Object Removal (MOR), a frame is successfully attacked if all designated target objects are undetected (i.e., for each targeted object, no detection has an IoU ¿ 0 with its ground-truth bounding box). For Targeted Object Removal (TOR), a frame is successfully attacked if the designated single target object has no detection with IoU ¿ 0 relative to its ground-truth bounding box and the number of false-positive detections does not exceed the threshold $\tau_{TOR}$. Here, we set the false-positive threshold $\tau_{TOR} = 2$.

Additionally, we introduce the Object Removal Rate (ORR) to measure the attack's effectiveness in reducing the number of detected objects relative to the ground truth. The ORR is defined as:

$$ORR = \frac{1}{N} \sum_{i=1}^{N} \max\left(0, 1 - \frac{|\mathcal{D}_i|}{|G_i|}\right),$$  (22)

where $|G_i|$ is the number of ground-truth objects in frame $i$,

and $|\mathcal{D}_i|$ is the number of detections output by the model in that frame. This metric ranges from 0 to 1 and represents the average proportion of ground-truth objects that are missed due to the attack.

**Implementation Details.** All four models were implemented using the PyTorch framework. For our proposed defense framework Cerberus, the weights for topological, semantic, and activation energy consistency were empirically set to $\lambda_1 = 0.3$, $\lambda_2 = 0.3$, and $\lambda_3 = 0.4$, respectively. The initial decision threshold was set to $\tau_0 = 0.23$, and the uncertainty parameter for adaptive threshold calibration was set to $\beta = 0.1$. To evaluate the robustness of the system, we employed BIM, PGD, and C&W attacks for both MOR and TOR scenarios. These attacks were executed within a unified iterative gradient optimization framework, with the maximum number of iterations set to 10 and an attack step size of 0.1. Furthermore, to assess the defense mechanism against a worst-case adversary, we constructed adaptive attacks. In this setup, the loss function explicitly incorporates defense-related constraints, simulating a strong, defense-aware attacker capable of bypassing superficial security measures.

**Qualitative evaluation of defense effectiveness.** Figure 8 provides a visual demonstration of the defense capabilities of Cerberus against MOR and TOR(random) attacks. As shown in the Figure 8, Cerberus achieves effective vehicle detection, which indicates its robustness against adversarial perturbations. Notably, the attack column reveals that the adversarial noise induces catastrophic degradation within the deep feature space as shown in rows 3 and 4, leading to a significant destruction of the feature maps. This observation aligns with our expectations. In contrast, Cerberus leverages multi-dimensional consistency checks within the feature space to identify and eliminate these anomalies, thereby effectively recovering the original feature maps. Vi-

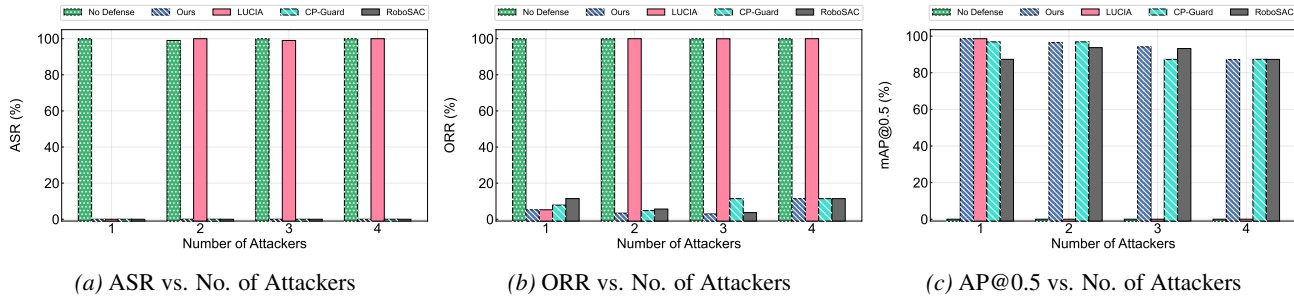

*(a)* ASR vs. No. of Attackers    *(b)* ORR vs. No. of Attackers    *(c)* AP@0.5 vs. No. of Attackers

*Figure 9.* Impact of the number of attackers. Our method demonstrates the strongest defense performance.

sually, the defended feature maps closely resemble the clean baseline, demonstrating that adversarial noise is successfully removed while preserving the benign environmental context. Both the perception results and the restored feature maps validate the effectiveness of `Cerberus` against such attacks.

**Impact of consistency verification across diverse attacks.** To rigorously validate the contribution of individual consistency constraints within `Cerberus`, we conducted a component-wise ablation study across four diverse collaborative perception backbones, as detailed in Table 5. We evaluated variants equipped solely with topological ($D_{topo}$), semantic ($D_{sem}$), or energy ($D_{mag}$) verifications against the full framework. The results reveal that relying on a single consistency dimension is insufficient for robust defense. Specifically, while structural consistency is fundamental, it is not a sufficient condition for security. This is starkly evident in the CoAlign and V2VAM models, where the $D_{topo}$-only variant suffers catastrophic failure under MOR attacks, yielding ORR exceeding 85% and ASR near 85%. This indicates that sophisticated adversaries can manipulate feature semantics while preserving the topological graph structure. However, by integrating semantic and energy constraints, the full `Cerberus` framework effectively rectifies these vulnerabilities, drastically reducing the ORR to 5.73% on CoAlign and 7.87% on V2VAM. Even on the AttFusion backbone, where single components perform relatively well, the full integration consistently achieves the lowest error rates (7.54% ORR) and highest detection precision compared to any individual metric. These findings confirm that topological, semantic, and energy consistencies provide complementary evidence, forming a comprehensive latent guardian that prevents attacks from exploiting any single feature dimension.

**Robustness to varying numbers of attackers.** We further evaluate the defense performance against varying numbers of attackers by increasing the attacker count from 1 to 4. As shown in Figure 9, baseline methods such as LUCIA show significant performance degradation as the number of attackers increases, with rising ORR and declining AP. In

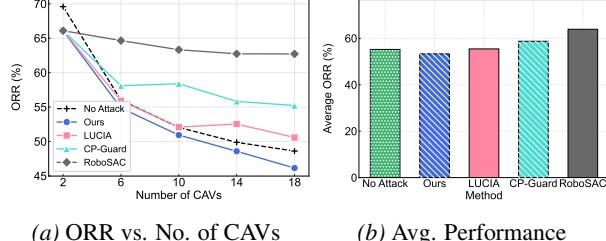

*(a)* ORR vs. No. of CAVs    *(b)* Avg. Performance

*Figure 10.* Impact of CAV numbers on the defense performance of `Cerberus` in a high-density traffic scenario (50 vehicles). Here we employ the CP model AttFusion under MOR attacks. (a) ORR vs. No. of CAVs. Our approach maintains superior stability and accuracy across varying collaboration scales compared to state-of-the-art methods. (b) Avg. Performance. Our method achieves the highest recovery rate, significantly outperforming other defenses.

contrast, `Cerberus` maintains strong stability. Even when facing four simultaneous attackers, it achieves a negligible ASR and the highest detection precision. This confirms that our multi-dimensional consistency checks can effectively identify malicious features regardless of the number of attackers, maintaining robust defense performance as attack intensity increases.

**Impact of CAV density.** To rigorously evaluate the scalability of `Cerberus` under extreme congestion, we follow LUCIA's (Wang et al., 2025) experimental protocol to construct a high-density traffic scenario containing 50 vehicles. In this complex setting, we vary the number of CAVs from 2 to 18 while launching MOR attacks. As illustrated in Figure 10(a), the no-attack baseline naturally decreases with the increasing number of CAVs, confirming that large-scale collaboration provides critical complementary perspectives to address occlusion issues. However, under attack conditions, conventional methods such as RoboSAC fail to scale effectively, exhibiting persistently high ORR due to the difficulty of distinguishing malicious features against the cluttered background of 50 vehicles. In contrast, `Cerberus` demonstrates superior robustness, closely adhering to the no-attack lower bound and consistently achieving the lowest ORR among all defense strategies. Figure 10(b) further corroborates this finding, showing that our method minimizes the

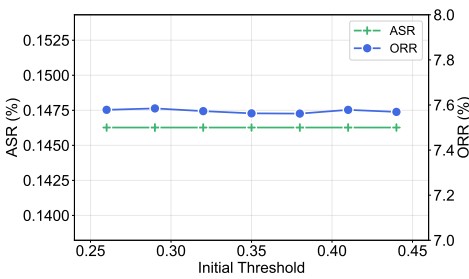

*Figure 11.* Impact of the initial threshold on `Cerberus` defense performance.

*Table 8.* Performance under different $T_{\max}$ settings.

| $T_{\max}$ | MOR-adaptive | MOR | TOR-adaptive | TOR |
|---|---|---|---|---|
| | ASR↓/ORR↓ | ASR↓/ORR↓ | ASR↓ | ASR↓ |
| 0.60 | 0.15% / 7.62% | 0.15% / 7.61% | 9.22% | 9.17% |
| 0.65 | 0.15% / 7.57% | 0.15% / 7.57% | 9.07% | 8.97% |
| 0.70 | 0.15% / 7.53% | 0.15% / 7.51% | 8.92% | 8.97% |
| 0.75 | 0.15% / 7.51% | 0.15% / 7.51% | 9.07% | 8.97% |

*Table 9.* Performance comparison under different $\lambda$ settings.

| Model | $(\lambda_1, \lambda_2, \lambda_3)$ | MOR | TOR |
|---|---|---|---|
| | | ASR/ORR↓ | ASR↓ |
| AttFusion | (0.33, 0.33, 0.34) | 0.15% / 7.58% | 9.07% |
| | (0.40, 0.40, 0.20) | 0.15% / 7.53% | 9.02% |
| | (0.50, 0.25, 0.25) | 0.20% / 7.99% | 9.31% |
| CoAlign | (0.33, 0.33, 0.34) | 0.05% / 5.89% | 6.78% |
| | (0.40, 0.40, 0.20) | 0.05% / 5.87% | 6.83% |
| | (0.50, 0.25, 0.25) | 0.05% / 6.05% | 6.87% |
| V2VAM | (0.33, 0.33, 0.34) | 0.24% / 8.38% | 9.90% |
| | (0.40, 0.40, 0.20) | 0.24% / 8.31% | 9.80% |
| | (0.50, 0.25, 0.25) | 0.29% / 8.66% | 10.04% |
| Where2comm | (0.33, 0.33, 0.34) | 0.10% / 4.55% | 5.12% |
| | (0.40, 0.40, 0.20) | 0.10% / 4.55% | 5.17% |
| | (0.50, 0.25, 0.25) | 0.10% / 4.56% | 5.22% |

*Table 10.* Performance comparison under different $\beta$ settings.

| $\beta$ | MOR-adaptive | MOR | TOR-adaptive | TOR |
|---|---|---|---|---|
| | ASR↓/ORR↓ | ASR↓/ORR↓ | ASR↓ | ASR↓ |
| 0.01 | 0.20% / 8.35% | 0.20% / 8.34% | 9.51% | 9.51% |
| 0.05 | 0.20% / 8.03% | 0.20% / 8.04% | 9.22% | 9.26% |
| 0.10 | 0.15% / 7.53% | 0.15% / 7.54% | 9.02% | 8.92% |
| 0.15 | 0.15% / 7.55% | 0.15% / 7.53% | 8.97% | 8.97% |

average ORR, thereby validating its effective exploitation of high-density collaboration even when the attack surface is significantly expanded.

**Sensitivity to initial threshold.** To assess the hyperparameter stability of our framework, we evaluated the defense performance while varying the initial threshold. As illustrated in Figure 11, both the ASR and ORR exhibit negligible fluctuations, maintaining consistently low error rates across the entire range. These results indicate that our defense mechanism does not rely on a precisely tuned optimal initialization. Instead, its stability stems from the subsequent uncertainty-aware adaptive adjustment, which dynamically calibrates the final decision boundary from different initial starting points. This robustness is advantageous for practical deployment, as it eliminates the need for laborious manual tuning or scenario-specific calibration.

**Sensitivity analysis of $T_{\max}$.** Table 8 demonstrates that our method is highly insensitive to the hyperparameter $T_{\max}$. Across both non-adaptive attacks (MOR/TOR) and adaptive attacks (MOR-adaptive/TOR-adaptive), the defense performance remains remarkably stable. For instance, as $T_{\max}$ increases from 0.60 to 0.75, the ASR for TOR-adaptive only fluctuates within a negligible range of roughly 0.3%, while the ORR for MOR-adaptive shows near-zero variance. This consistency proves that our defense capability does not rely on meticulous parameter tuning, ensuring strong robustness and ease of deployment in diverse and unpredictable environments.

**Sensitivity analysis of $\lambda$ settings.** The results in Table 9 indicate that our proposed method is remarkably robust to

the variations of weight parameters $(\lambda_1, \lambda_2, \lambda_3)$. We tested several typical combinations, including balanced weights and skewed distributions. Across all four evaluated models, the ASR and ORR remain highly consistent with only marginal fluctuations. This stability demonstrates that the effectiveness of our defense is not sensitive to the specific weighting of different components, further validating the reliability and generalizability of our method across various collaborative perception frameworks.

**Sensitivity analysis of $\beta$ settings.** Table 10 presents the performance comparison under different $\beta$ settings. The results show that varying $\beta$ has a certain influence on the defense performance. Specifically, as $\beta$ increases, the ASR and ORR values fluctuate slightly across different attack settings, indicating that the balance controlled by $\beta$ affects the behavior of the defense mechanism. Nevertheless, the overall performance remains relatively stable, and no significant degradation is observed. In particular, the defense consistently maintains low ASR values under both MOR and TOR attacks, demonstrating that the proposed method is not overly sensitive to the choice of $\beta$. These results indicate that although the parameter setting can influence the defense effectiveness to some extent, the proposed method remains robust and does not fail under different $\beta$ configurations.

**Grouping strategy analysis.** To validate the effectiveness of our grouping strategy, we compare it against random grouping across multiple scene types. We adopt two eval-

*Table 11.* Our grouping vs. random grouping.

| Grouping | Scene | Visibility Overlap ↑ | One-sided Object Rate ↓ |
|---|---|---|---|
| Ours | All | 0.7724 | 0.2276 |
| Random | All | 0.7512 | 0.2488 |
| Ours | Dense | 0.7509 | 0.2491 |
| Random | Dense | 0.7505 | 0.2495 |
| Ours | Asymmetric | 0.8097 | 0.1903 |
| Random | Asymmetric | 0.7481 | 0.2519 |

uation metrics: visibility overlap, measured as the IoU between the object-ID sets observed by the two groups, and one-sided object rate, defined as the fraction of objects visible to only one group. A higher visibility overlap and a lower one-sided object rate indicate more balanced grouping with less viewpoint-induced bias. As shown in Table 11, our grouping consistently outperforms random grouping across all scene types. On the full set, our method achieves a visibility overlap of 0.7724 and a one-sided object rate of 0.2276, compared to 0.7512 and 0.2488 for random grouping. The advantage is most pronounced in asymmetric scenes, where our method yields a visibility overlap of 0.8097 versus 0.7481, representing the largest margin across all subsets. These results confirm that our grouping strategy produces more balanced partitions, particularly in challenging asymmetric configurations.

**Defense latency analysis.** We report the end-to-end defense latency under varying numbers of CAVs in Table 12. Our method consistently achieves the lowest latency across all settings. With 40 CAVs, our method runs in 259.72 ms, compared to 775.75 ms for Robosac and 2920.73 ms for CP-Guard, achieving a speedup of approximately $3 \times$ and $11\times$, respectively. Notably, CP-Guard exhibits severe scalability issues, with latency growing super-linearly as the number of CAVs increases, rendering it impractical in large-scale deployment scenarios. In contrast, our method scales gracefully, maintaining reasonable latency even at $N$=40.

*Table 12.* End-to-end defense latency (ms).

| Method | $N$=5 | $N$=10 | $N$=15 | $N$=20 | $N$=30 | $N$=40 |
|---|---|---|---|---|---|---|
| Ours | 67.71 | 107.95 | 124.68 | 147.68 | 195.09 | 259.72 |
| Robosac | 112.61 | 193.51 | 270.65 | 350.75 | 530.35 | 775.75 |
| CP-Guard | 231.35 | 572.76 | 799.85 | 1203.12 | 1715.92 | 2920.73 |

**Detection basis analysis.** Table 13 validates the rationale behind our three-dimensional detection mechanism by comparing the topological, semantic, and energy features of benign vehicles, attackers before attack, and attackers after attack. Before launching an attack, the attacker's feature statistics closely resemble those of benign vehicles across all three dimensions, confirming that our detection does not rely on identity-level priors. Once the attack is activated, however, all three features exhibit pronounced and consis-

*Table 13.* Adaptive white-box attack performance.

| Group | Anomaly Score ↓ | Topological ↑ | Semantic ↑ | Energy ↓ |
|---|---|---|---|---|
| Benign | 0.2070 ± 0.0337 | 0.6761 ± 0.0715 | 0.9201 ± 0.0075 | 0.2146 ± 0.0379 |
| Attacker (before attack) | 0.2045 ± 0.0337 | 0.6818 ± 0.0740 | 0.9217 ± 0.0082 | 0.2138 ± 0.0358 |
| Attacker (after attack) | 0.4613 ± 0.0128 | 0.6160 ± 0.0288 | 0.5042 ± 0.0011 | 0.4934 ± 0.0141 |

*Table 14.* Verification count: our grouping vs. per-vehicle.

| Setting | Mode | Avg. Ver. per Frame | Avg. Single-Vehicle Ver. per Frame |
|---|---|---|---|
| No Attack | Our Grouping | 2.00 | 0 |
| | Direct Per-Vehicle | 5.63 | 5.63 |
| MOR | Our Grouping | 4.47 | 2.04 |
| | Direct Per-Vehicle | 5.63 | 5.63 |
| TOR | Our Grouping | 4.47 | 2.04 |
| | Direct Per-Vehicle | 5.63 | 5.63 |

tent shifts: the anomaly score rises sharply, the semantic similarity drops substantially, and the energy increases significantly, while the topological consistency also degrades. These results demonstrate that adversarial manipulation inevitably leaves detectable traces in the feature space, providing a strong empirical foundation for our multi-dimensional detection approach.

**Verification efficiency analysis.** To demonstrate the efficiency advantage of our grouping strategy, we compare the average number of verifications per frame between our grouping-based approach and direct per-vehicle comparison, as shown in Table 14. Direct per-vehicle verification requires checking every CAV individually, resulting in a constant $O(N)$ overhead regardless of the attack scenario. In contrast, our approach performs a global group-level check first, and only initiates fine-grained binary search when an anomaly is detected, reducing the overall complexity to $O(\log N)$. In the no-attack scenario, our method requires only 2 group-level verifications per frame with zero single-vehicle checks, compared to 5.63 individual verifications for the direct approach. Under both MOR and TOR attack scenarios, our method increases to 4.47 total verifications with 2.04 single-vehicle checks per frame, still substantially fewer than the 5.63 required by direct per-vehicle comparison. These results confirm that our grouping strategy significantly reduces verification overhead while preserving detection capability.

**Robustness against persistent adversaries.** We further evaluate our defense against persistent adversaries, where the attacker continuously attempts to manipulate perception results over multiple frames before launching a full-scale attack. As shown in Table 15, our defense maintains consistently low attack success rates across all CP backbones. This robustness stems from two key properties of our de-

*Table 15.* Robustness against persistent adversaries.

| Model | MOR | | TOR |
|---|---|---|---|
| | ORR↓ | ASR↓ | ASR↓ |
| AttFusion | 4.55% | 0.04% | 4.60% |
| CoAlign | 2.18% | 0% | 0% |
| Where2comm | 6.18% | 0.09% | 6.97% |
| V2VAM | 10.85% | 0.39% | 20.52% |

*Table 17.* Extended results on V2V4Real.

| Model | Defense | ASR ↓ | ORR ↓ |
|---|---|---|---|
| AttFusion | No Defense | 44.26% | 88.52% |
| | Our `Cerberus` | 9.05% | 29.88% |
| V2XViT | No Defense | 16.03% | 63.41% |
| | Our `Cerberus` | 10.13% | 30.11% |

sign. First, our multi-dimensional orthogonal verification ensures that any sustained manipulation inevitably introduces detectable inconsistencies across structural and semantic features. Second, our adaptive mechanism enforces a strict global maximum threshold as an absolute ceiling, preventing accumulated perturbations from gradually shifting the decision boundary. As a result, persistent attack attempts gain no meaningful advantage over single-frame attacks, confirming that our defense remains effective against adversaries that operate over extended time horizons.

**Comparison with lazy verification.** To justify our grouping strategy over a coarse-to-fine alternative, we consider a lazy verification baseline that first fuses all received feature maps into a single global representation and performs a single global validation. Only when an anomaly is detected does the system proceed to grouping and recursive localization. We compare detection recall of this lazy verification approach against our method under PGD attacks of varying strengths with 11 collaborating vehicles, as shown in Table 16. When the attack strength is low, lazy verification fails to detect any attacks due to the anomaly dilution effect, where malicious features are suppressed during the fusion process before the global check is triggered. In contrast, our grouping method maintains a consistently high detection recall of 80.43% across all tested values of $\epsilon$. As attack strength increases, lazy verification gradually recovers some detection capability, but still falls significantly behind our method. These results demonstrate that a single global validation is fundamentally insufficient for reliable detection under weak attacks, and confirm the necessity of performing grouping from the outset rather than deferring it as a conditional fallback.

*Table 16.* Comparison between our grouping strategy and lazy verification under different attack intensities $\epsilon$.

| $\epsilon$ | Detection Recall ↑ | |
|---|---|---|
| | Ours | Lazy Verification |
| 0.030 | 80.43% | 0.00% |
| 0.034 | 80.43% | 0.00% |
| 0.038 | 80.43% | 2.17% |
| 0.042 | 80.43% | 26.09% |

**Results on V2V4Real.** Table 17 reports the defense performance against MOR attacks on the V2V4Real dataset across

two CP backbones. Without any defense, both models exhibit high attack success rates, with AttFusion reaching an ASR of 44.26% and an ORR of 88.52%, indicating significant vulnerability to malicious interference. After applying our defense, ASR and ORR are substantially reduced across both backbones. For AttFusion, ASR drops from 44.26% to 9.05% and ORR from 88.52% to 29.88%. For V2XViT, similar reductions are observed. These results demonstrate that our defense generalizes well to real-world data and remains effective across different CP architectures.

