# OpenReview forum: "The Latent Guardian: Defending Collaborative Perception via Feature-Level Consistency Verification"
_ICML.cc/2026/Conference — ICML 2026 regular_

### Official Review · Reviewer_VCe5 · 2026-03-10

**Soundness:** 3
**Presentation:** 3
**Significance:** 3
**Originality:** 2
**Overall Recommendation:** 4
**Confidence:** 4

**Summary:**

This paper proposes Cerberus, a defense framework for collaborative perception systems against adversarial attacks, which detects malicious vehicles by shifting from output-level consensus to consistency verification in the latent feature space. The method employs isomorphic view grouping and multi-dimensional feature consistency checks (topology, semantics, and energy) to identify anomalous features, and further introduces an uncertainty-aware mechanism to dynamically adjust detection thresholds, significantly improving system robustness across multiple models and attack scenarios.

**Compliance With Llm Reviewing Policy:**

Affirmed.

**Final Justification:**

The additional experimental results provided by the authors sufficiently validate the effectiveness of the proposed method. Defending Collaborative Perception is meaningful for the deployment of collaborative perception systems. Therefore, I have increased my score.

**Key Questions For Authors:**

**Questions:**

- The current threat model assumes a single malicious vehicle. How would the proposed method perform if multiple malicious vehicles collude and are assigned to different groups during the grouping process?

- The adaptive threshold depends on several hyperparameters (e.g., $\beta$ and $T_{max}$). Could the authors provide more insight into how sensitive the method is to these parameters?

**Limitations:**

The authors could strengthen the paper by explicitly discussing potential limitations such as scalability, robustness under stronger attack models, and computational overhead in real-time settings.

**Strengths And Weaknesses:**

**Strengths:**

- This paper addresses an important security problem in collaborative perception for autonomous driving and introduces a feature-level consistency verification framework with a clear design motivation.

- Extensive experiments across multiple models, attack scenarios, and real-world tests.


**Weaknesses:**

- The proposed isomorphic view grouping relies on angular sorting and interleaved sampling of neighboring vehicles. In dense or highly dynamic traffic scenarios, maintaining such grouping may incur additional overhead. Moreover, when only a small number of collaborators are available (e.g., 2–3 vehicles), the grouping strategy may lead to sparse groups, potentially degrading the quality of fused perception.

- The defense operates in the latent feature space and requires multiple consistency checks, including SSIM, cosine similarity, and feature magnitude comparisons. This multi-dimensional verification may introduce additional inference latency, and the paper provides limited quantitative analysis of the computational overhead in real-time autonomous driving settings.

---

> ### Author Rebuttal · Authors · 2026-03-29
>
> [W1]For isomorphic view grouping, we respond from two aspects. First, regarding computational overhead, the angular sorting and interleaved sampling operate purely on lightweight 1D metadata with a time complexity of $O(N \log N)$, where $N$ is the number of connected vehicles. When executed on the CPU, it requires negligible time compared to deep neural network inference. Moreover, in most practical V2V settings, $N$ is typically small (e.g., $N \le 10$), making the overhead even lower. Table 1 also confirms that our method incurs lower latency than all baselines. Second, regarding sparse groups, there is a slight misunderstanding of our pipeline. The grouping strategy is strictly an intermediate step for anomaly detection. Once malicious nodes are filtered out, all verified benign vehicles are merged back for the final perception fusion. Therefore, even with only 2–3 collaborators, no valid benign features are discarded, avoiding negative impact on the final perception quality.
>
> [W2] We supplement experiments on the per-frame processing latency after deploying our defense under different numbers of CAVs (including the final inference stage), as well as the latency of each component. Tables 1 and 2 show the corresponding results.
>
> End-to-end Efficiency (Table 1): Our method consistently incurs lower latency than other methods, indicating that it introduces only controllable overhead. Even in an extreme scenario with 40 vehicles, our overhead (259.72 ms) is still much lower than that of CP-Guard (2920.73 ms).
>
> Component Overhead (Table 2): The results show that the latency of each component in our defense is small and only increases when the number of vehicles becomes large. Notably, the Topological and Energy checks maintain a near-constant, negligible latency regardless of the CAV count.
>
> Considering that the number of collaborating vehicles in real-world scenarios is usually small to moderate, these results demonstrate that our method is highly efficient.
>
> *Table 1. End-to-end defense latency  (ms)*
> | Method | CAV=5 | CAV=10 | CAV=15 | CAV=20 | CAV=30 | CAV=40 |
> | --- | --- | --- | --- | --- | --- | --- |
> | Ours | 67.71 | 107.95 | 124.68 | 147.68 | 195.09 | 259.72 |
> | Robosac (ICCV'23) | 112.61 | 193.51 | 270.65 | 350.75 | 530.35 | 775.75 |
> | CP-Guard (AAAI'25) | 231.35 | 572.76 | 799.85 | 1203.12 | 1715.92 | 2920.73 |
>
> *Table 2. Latency of each component in our defense (ms)*
> | Component | CAV=5 | CAV=10 | CAV=15 | CAV=20 | CAV=30 | CAV=40 |
> | --- | --- | --- | --- | --- | --- | --- |
> | Grouping | 0.86 | 3.08 | 6.64 | 11.09 | 20.74 | 30.97 |
> | Topological | 1.66 | 2.36 | 2.55 | 2.62 | 2.58 | 2.65 |
> | Semantic | 19.48 | 28.15 | 30.76 | 32.00 | 33.14 | 34.29 |
> | Energy | 0.88 | 1.22 | 1.37 | 1.43 | 1.32 | 1.34 |
>
> [Q1]We sincerely apologize for the confusion. *Our defense method can effectively defend against attacks regardless of whether there is only one attacker or multiple attackers launching attacks simultaneously, even when malicious vehicles are present in both groups.* Specifically, in our method, all vehicles are first divided into two groups and evaluated separately. If a group is assessed as benign, all vehicles in that group are directly regarded as benign and are allowed to participate in collaborative perception. If a group is detected as malicious, that group is further divided into two subgroups for another round of evaluation. This process is repeated iteratively until all benign vehicles are identified. *Furthermore, as demonstrated in Fig. 9 of the manuscript, our method consistently outperforms existing defenses when facing multiple simultaneous attackers.*
>
> [Q2] We have added experiments on $\beta$, $T_{\max}$, and $\lambda_{1,2,3}$. The results show that our method exhibits low sensitivity to these parameters. The sensitivity testing of $T_{max}$  on AttFusion is shown in the table below. It is evident that as $T_{max}$  varies, our method maintains effective defense performance against both MOR/TOR sombra (USENIX Sec'25) and the adaptive attack MOR/TOR-adaptive which possesses full knowledge of the defense mechanism. Regarding the analysis of parameters $\lambda_{1,2,3}$, and $\beta$, please refer to our response [W4] to Reviewer 4NSZ due to space constraints.
>
> *Table 3. Effect of $T_{\max}$ on defense performance under different attacks.*
> | $T_{\max}$ | MOR-adaptive (ASR/ORR) ↓ | MOR-sombra (ASR/ORR) ↓ | TOR-adaptive (ASR) ↓ | TOR-sombra (ASR) ↓ |
> |:----------:|:--------------------------:|:----------------------:|:----------------------:|:------------------:|
> | 0.60 | 0.15% / 7.62% | 0.15% / 7.61% | 9.22% | 9.17% |
> | 0.65 | 0.15% / 7.57% | 0.15% / 7.57% | 9.07% | 8.97% |
> | 0.70 | 0.15% / 7.53% | 0.15% / 7.51% | 8.92% | 8.97% |
> | 0.75 | 0.15% / 7.51% | 0.15% / 7.51% | 9.07% | 8.97% |

---

> > ### Author Rebuttal · Reviewer_VCe5 · 2026-04-03
> >
> > Thanks to the author for the reply and clarification.

---

> > > ### Author Response · Authors · 2026-04-06
> > >
> > > Thank you very much for your time and effort. If you have any further comments or questions, please feel free to let us know. If our responses have addressed your concerns, we would also appreciate it if you could reconsider your evaluation of our paper. Thank you again for your constructive feedback.

---

### Official Review · Reviewer_7TGc · 2026-03-12

**Soundness:** 2
**Presentation:** 3
**Significance:** 2
**Originality:** 3
**Overall Recommendation:** 3
**Confidence:** 3

**Summary:**

The paper studies adversarial robustness for intermediate-fusion collaborative perception (CP), where neighboring vehicles share feature maps rather than raw sensor data or final detections. The authors argue that existing defenses focus too much on output-level consensus and therefore miss stealthy attacks that can keep outputs plausible while corrupting latent features.
The proposed method, Cerberus, has three parts:

1. Isomorphic View Fusion, which sorts collaborators by azimuth and interleaves them into two groups to reduce viewpoint mismatch;
2. Feature-Level Consistency Verification, which scores discrepancies between group features and ego features using topological similarity (SSIM-like), semantic direction (cosine dissimilarity), and energy shift (L1-based magnitude discrepancy);
3. Uncertainty-Aware Adaptive Decision, which sets a dynamic threshold based on ego-feature entropy plus a sliding-window estimate of benign score statistics.
Experiments are run on OPV2V using four fusion backbones and multiple attacks, including MOR, TOR variants, BIM, PGD, C&W, and a defense-aware adaptive attacker. The headline numbers are strong in several settings; for example, under MOR on V2VAM, Cerberus reports ORR/ASR of 7.87%/0.19% and under adaptive attacks it still reports relatively strong performance.

**Compliance With Llm Reviewing Policy:**

Affirmed.

**Key Questions For Authors:**

1. How strong is the adaptive attacker really? Did the authors try more optimization steps, multiple restarts, or attacks that directly optimize through the recursive group partitioning/localization procedure rather than only the joint anomaly score?

2. Can the authors justify the statement that attackers cannot produce benign-looking features under a white-box setup? Right now this reads more like an intuition than a demonstrated property.

3. How often does azimuth-interleaving fail to produce genuinely comparable groups in sparse or asymmetric scenes? A diagnostic analysis would be very helpful.

**Strengths And Weaknesses:**

Defending collaborative perception against malicious participants is a real weakness of connected driving systems, and focusing on intermediate-fusion attacks is practically relevant because this is exactly where modern CP systems often operate.

The paper has a clear intuition: subtle attacks may be easier to spot in feature space than after decoding, and moving the defense earlier in the pipeline is a sensible direction. The page-2 figure and method overview make that intuition easy to follow.

The experimental coverage is broader than average for this topic: multiple backbones, multiple attacks, an ablation, an adaptive-attacker experiment, an efficiency plot, and a small real-world testbed.

Weakness:

1. The paper’s central assumption is stronger than the evidence: it states that attackers can forge outputs but “cannot generate feature maps that closely resemble those of benign vehicles.” I do not think the paper really establishes this, especially under a white-box threat model. In practice, the defense itself is built from differentiable or near-differentiable quantities (SSIM-like structure, cosine similarity, L1 magnitude, adaptive thresholding), so the claim that benign-looking features are hard to construct needs more justification than the paper provides.

2. The isomorphic grouping step feels heuristic. Sorting collaborators by azimuth and assigning alternate vehicles into two groups is a clever trick, but I do not see a strong reason to believe this produces genuinely comparable views in complex scenes. Traffic density is uneven, visibility is asymmetric, range varies, and occlusion depends on far more than angle. The paper presents this as removing viewpoint bias, but in many scenes it may only reduce it loosely.

3. Some of the empirical claims are stronger than the table supports. The paper says Cerberus “consistently achieves low ASR across all settings,” but under TOR(Out) the reported ASR remains 32.62% for AttFusion, 26.81% for CoAlign, and 38.22% for V2VAM. Those are meaningful reductions over baselines, but I would not call them uniformly low.

---

> ### Author Rebuttal · Authors · 2026-03-29
>
> [W1&Q2] Your concern is very important. We would like to clarify that our claim is not that forging benign-looking features is theoretically impossible, but rather that it is not achieved in practice. Although Cerberus's components are differentiable, the benign reference feature maps derive from live, attacker-uncontrollable sensor measurements varying dynamically across frames and vehicles. An attacker must simultaneously satisfy three constraints against references unobservable in advance, producing conflicting gradient signals that make joint optimization fundamentally harder than attacking a static boundary. Additionally, we evaluate five attacks from USENIX Security '25, AAAI '25, and ICCV '23, and find that none of them can produce feature maps indistinguishable from those of benign agents. Furthermore, we evaluate the capability of adaptive white-box attacks to forge features. As shown in Table 1, they cannot simultaneously achieve performance comparable to benign vehicles in three aspects, topological, semantic, and energy, indicating that it is difficult to forge feature maps similar to those of benign vehicles.
>
> Table 1. Adaptive white-box attack performance
> | Group | Anomaly Score ↓ | Topological ↑ | Semantic ↑ | Energy ↓ |
> |---|---:|---:|---:|---:|
> | Benign | 0.2070 ± 0.0337 | 0.6761 ± 0.0715 | 0.9201 ± 0.0075 | 0.2146 ± 0.0379 |
> | Attacker (before attack) | 0.2045 ± 0.0337 | 0.6818 ± 0.0740 | 0.9217 ± 0.0082 | 0.2138 ± 0.0358 |
> | Attacker (after attack) | 0.4613 ± 0.0128 | 0.6160 ± 0.0288 | 0.5042 ± 0.0011 | 0.4934 ± 0.0141 |
>
> [W2&Q3] We agree that in highly asymmetric scenes, our grouping strategy mitigates rather than perfectly eliminates viewpoint bias. However, it is not intended to construct identical views. Its sole purpose is to establish a statistically comparable baseline for anomaly detection. By broadly aligning spatial coverage, macroscopic feature distributions become comparable enough to expose malicious deviations. Crucially, any residual variance caused by natural occlusions or uneven density is inherently absorbed by our adaptive thresholding, preventing false positives while keeping the defense highly lightweight. The experiments shown in the newly added Table 2 also support this conclusion. As shown in Table 2, in sparse and asymmetric scenarios, the failure rates of our grouping are 8.12% and 33.31%, respectively. However, the clean false positive rate remains 0, indicating that, in the absence of attacks, our method does not misclassify benign vehicles as malicious. The failure rate is defined as the percentage of frames where the difference between the two groups exceeds the upper 10% boundary of random balanced splits with the same number of collaborators. Sparse denotes clean test scenes with at most three candidate collaborators, and asymmetric denotes the 25% of clean traffic-jam scenes with the most uneven collaborator layouts around the ego vehicle.
>
> *Table 2. Impact of grouping*
> | Scenario   | Inter-group Similarity | Failure Rate (%) | Clean FPR (%) |
> |------------|--------------|------------------|---------------|
> | Sparse     | 0.05    | 8.12           | 0   |
> | Asymmetric | 0.02    | 33.32          | 0   |
>
> [W3] We apologize for the misunderstanding. We will revise this section in the modified manuscript. Our primary intention was to demonstrate that our proposed method achieves the lowest attack success rate among all evaluated defense mechanisms.
>
> [Q1] The adaptive attacker in the paper has full knowledge of our defense and directly optimizes by combining anomaly scores from all three dimensions. In this case, our defense still demonstrates significant robustness. Additionally, we conducted new experiments. We significantly strengthened the adaptive attacker by employing more optimization steps, multiple random restarts, and explicitly optimizing through our recursive group partitioning procedure. The results shown in the Table 3 below demonstrate that, despite these advanced, structure-aware optimizations, Cerberus consistently maintains robust detection and localization performance.
>
> *Table 3. Robustness against adaptive attacks under different settings*
> | Variant | Surrogate | Iter. | Random Starts | MOR ASR / ORR (%) | TOR ASR (%) | MOR AP@0.5 / AP@0.7 | TOR AP@0.5 / AP@0.7 |
> | --- | --- | ---: | ---: | ---: | ---: | ---: | ---: |
> | V0 original joint surrogate | joint | 10 | 1 | 0.15 / 7.53 | 9.02 | 0.84 / 0.72 | 0.84 / 0.72 |
> | V1 more optimization steps | joint | 20 | 1 | 0.20 / 8.04 | 9.17 | 0.84 / 0.72 | 0.84 / 0.72 |
> | V2 joint + 5 random restarts | joint | 20 | 5 | 0.20 / 8.05 | 9.18 | 0.84 / 0.72 | 0.84 / 0.72 |
> | V3 grouping-aware recursive surrogate | grouping-aware | 20 | 1 | 0.20 / 8.06 | 9.17  | 0.84 / 0.72 | 0.84 / 0.72 |
> | V4 grouping-aware + 5 random restarts | grouping-aware | 20 | 5 | 0.20 / 8.05 | 9.18 | 0.83 / 0.71 | 0.83 / 0.71 |

---

> > ### Author Rebuttal · Reviewer_7TGc · 2026-04-05
> >
> > I appreciate the rebuttal from the authors, the authors have resolved most of my solvable concerns, but I still think the overall reduction is not convincing enough to address the adversarial setting.

---

> > > ### Author Response · Authors · 2026-04-07
> > >
> > > Thank you very much for your valuable comments and response. We sincerely apologize for any misunderstanding caused by our previous reply. As shown in Table 1, our method achieves the best defense performance compared with existing state-of-the-art methods. Regarding your concern that, under the TOR (Out) attack, our defense only reduces the ASR to around 30%, this is because the attack target is located outside the ego vehicle’s field of view, making this type of attack difficult for all defense methods to handle effectively. **Nevertheless, it is worth noting that our method still achieves the best defense performance under this attack setting.**
> > >
> > > **Table 1. ASR↓ after deploying the defense method under various attack**
> > >
> > > | Attack | Model | No Defense | ROBOSAC (ICCV'23) | LUCIA (USENIX Sec.'25) | CP-Guard (AAAI'25) | Ours |
> > > |---|---|---|---|---|---|---|
> > > | TOR (In) | AttFusion | 99.61% | 96.34% | 6.05% | 97.42% | **4.19%** |
> > > | TOR (In) | CoAlign | 99.12% | 96.83% | 5.07% | 96.64% | **2.97%** |
> > > | TOR (In) | Where2comm | 78.21% | 18.33% | 5.31% | 19.89% | **3.31%** |
> > > | TOR (In) | V2VAM | 97.37% | 92.89% | 14.14% | 93.95% | **4.48%** |
> > > | TOR (Out) | AttFusion | 98.79% | 96.78% | 47.25% | 97.32% | **32.62%** |
> > > | TOR (Out) | CoAlign | 98.39% | 98.49% | 40.03% | 96.64% | **26.81%** |
> > > | TOR (Out) | Where2comm | 77.76% | 58.81% | 28.08% | 55.29% | **15.26%** |
> > > | TOR (Out) | V2VAM | 97.22% | 97.56% | 62.32% | 94.83% | **38.22%** |
> > > | TOR (Random) | AttFusion | 98.09% | 95.90% | 9.02% | 95.32% | **8.92%** |
> > > | TOR (Random) | CoAlign | 97.22% | 95.07% | 6.73% | 95.56% | **6.73%** |
> > > | TOR (Random) | Where2comm | 58.56% | 16.86% | 5.52% | 22.38% | **5.16%** |
> > > | TOR (Random) | V2VAM | 94.93% | 89.62% | 20.33% | 91.91% | **9.56%** |
> > >
> > > Specifically, **under the TOR (Out) scenario, our method further reduces the ASR by an average of approximately 16.19% compared with the state-of-the-art baseline.** The improvement is most significant on V2VAM, where the ASR is reduced by 24.10%, while the gain on Where2comm is relatively smaller but still reaches 12.82%. Overall, our method demonstrates clearly stronger defense capability against TOR (Out) attacks than the existing state-of-the-art method.
> > >
> > > Thank you very much for the time and effort you devoted to reviewing our manuscript. We hope that our response has addressed your concerns.

---

### Official Review · Reviewer_4NSZ · 2026-03-12

**Soundness:** 3
**Presentation:** 3
**Significance:** 3
**Originality:** 3
**Overall Recommendation:** 4
**Confidence:** 4

**Summary:**

This work proposes Cerberus, a defense framework designed to counter adversarial attacks in collaborative perception systems by leveraging multi-dimensional consistency within the feature space.

**Compliance With Llm Reviewing Policy:**

Affirmed.

**Key Questions For Authors:**

None

**Limitations:**

Yes

**Strengths And Weaknesses:**

Strength:
1. This work designs an isomorphic view fusion module to eliminate geometric viewpoints bias via interleaved sampling, a multi-dimensional metric to detect adversarial attacks, and an uncertainty-aware adaptive decision module to integrate perceptual entropy to quantify environmental ambiguity.
2. Overall the paper is well written, the experiments are comprehensive with a variety of settings and ablation analyses.

Weakness:
1. Metrics like SSIM, cosine similarity and L1 distance are common, the paper fails to clarify how these standard metrics are customized for collaborative perception’s specific pain points, making innovations seem like simple combinations.
2. Experiments rely solely on a single simulated dataset OPV2V, which is insufficiently convincing. Evaluation on real-world public dataset (e.g., V2X-Real, V2V4Real, DAIR-V2X) are needed.
3. The evaluation metrics include mAP, however, the OPV2V dataset contains only a single category “vehicle”. Were other object classes are involved in the experiments?
4. Critical parameters (e.g., the balancing coefficients \lambda_{1,2,3}, the uncertainty parameter \beta) lack ablation tests. There is no proof of their optimality, weakening the method’s technical rigor.
5. Lack of ablation tests of the proposed sorting-based grouping strategy, failing to demonstrate its advantages against normal methods (e.g. random grouping).

---

> ### Author Rebuttal · Authors · 2026-03-30
>
> [W1] Our innovation lies not in inventing metrics, but mapping them to specific V2X attack vectors: SSIM detects topological manipulations (e.g., fake vehicles), Cosine monitors semantic shifts, and L1 restricts abnormal energy inflation. Enforcing this joint multi-dimensional constraint simultaneously creates an intractable optimization trade-off for attackers, far beyond a simple combination. The results in Table 1 of our response to Reviewer 7TGc also support this point.
>
> [W2] We conducted experiments on the V2V4Real dataset. The results for MOR attacks in Table 1 show that our method can effectively defend against the attacks. Meanwhile, the real-world evaluation in Section 5.5 of the manuscript further demonstrates the applicability of the proposed defense in real-world settings.
>
> *Table 1. Extended results on V2V4Real*
> | Model            | Defense    | ASR  | ORR    |
> |--|--|--:|--:|
> | AttFusion  | No Defense | 44.26% | 88.52% |
> | AttFusion  | Cerberus   | 9.05%  | 29.88% |
> | V2XViT           | No Defense | 16.03% | 63.41% |
> | V2XViT           | Cerberus   | 10.13% | 30.11% |
>
>
> [W3] We only utilized the vehicle category. We thank the reviewer for pointing out this terminology inaccuracy. The use of mAP was intended to maintain consistency with other papers using this dataset, such as LUCIA (USENIX Sec 2025). We will correct this in the revised manuscript.
>
> [W4] For the key parameters, we have conducted additional ablation experiments for $\lambda_{1,2,3}$ and $\beta$. As shown in the table below, the results demonstrate that Cerberus maintains stable defense performance across different parameter settings. This confirms the framework's robustness to parameter variations, proving that our method does not rely on a strictly optimal configuration.
>
> *Table 2. Performance comparison under different $\lambda$ settings*
> | Model           | $(\lambda_1, \lambda_2, \lambda_3)$ | MOR (ASR/ORR) | TOR (ASR) |
> |:----------------|:-----------------------------------:|:-------------:|:---------:|
> | AttFusion | (0.33, 0.33, 0.34)                  | 0.15% / 7.58% | 9.07%     |
> |                 | (0.40, 0.40, 0.20)                  | 0.15% / 7.53% | 9.02%        |
> |                 | (0.50, 0.25, 0.25)                  | 0.20% / 7.99% | 9.31%     |
> | CoAlign         | (0.33, 0.33, 0.34)                  | 0.05% / 5.89% | 6.78%     |
> |                 | (0.40, 0.40, 0.20)                  | 0.05% / 5.87% | 6.83%     |
> |                 | (0.50, 0.25, 0.25)                  | 0.05% / 6.05% | 6.87%     |
> | V2VAM           | (0.33, 0.33, 0.34)                  | 0.24% / 8.38% | 9.90%  |
> |                 | (0.40, 0.40, 0.20)                  | 0.24% / 8.31% | 9.80%     |
> |                 | (0.50, 0.25, 0.25)                  | 0.29% / 8.66% | 10.04%    |
> | Where2comm      | (0.33, 0.33, 0.34)                  | 0.10% / 4.55% | 5.12%   |
> |                 | (0.40, 0.40, 0.20)                  | 0.10% / 4.55% | 5.17%     |
> |                 | (0.50, 0.25, 0.25)                  | 0.10% / 4.56% | 5.22%     |
>
> *Table 3. Performance comparison under different $\beta$ settings*
> | Attack | Metric | $\beta=0.01$ | $\beta=0.05$ | $\beta=0.10$ | $\beta=0.15$ |
> | --- | --- | --- | --- | --- | --- |
> | MOR-adaptive | ASR / ORR | 0.20% / 8.35% | 0.20% / 8.03% | 0.15% / 7.53% | 0.15% / 7.55% |
> | MOR-sombra | ASR / ORR | 0.20% / 8.34% | 0.20% / 8.04% | 0.15% / 7.54% | 0.15% / 7.53% |
> | TOR-adaptive | ASR | 9.51% | 9.22% | 9.02% | 8.97% |
> | TOR-sombra | ASR | 9.51% | 9.26% | 8.92% | 8.97% |
>
> [W5] We evaluate the vehicle partitioning capability of our grouping strategy against random grouping in clean scenes (including dense and asymmetric scenarios). We use two metrics: Visibility Overlap, defined as the IoU between the object-ID sets observed by the two groups, and One-sided Object Rate, defined as the fraction of objects visible to only one group. Higher visibility overlap and lower one-sided rate indicate a more balanced grouping with less viewpoint-induced bias. As shown below, our grouping consistently outperforms random grouping on the full set, with the largest gain on the asymmetric subset. This shows that our grouping is not arbitrary: it produces more balanced partitions than a normal baseline, especially in asymmetric scenes.
>
> *Table 4. Our grouping vs. random grouping*
> | Grouping | Scene |  Visibility Overlap ↑ | One-sided Object Rate ↓ |
> |:---------|:----------------------|:---------------|:----------------------|
> | Ours     | all                   | 0.7724         | 0.2276         |
> | Random   | all                   | 0.7512         | 0.2488     |
> | Ours     | dense                 | 0.7509         | 0.2491    |
> | Random   | dense                 | 0.7505         | 0.2495    |
> | Ours     | asymmetric         | 0.8097         | 0.1903         |
> | Random   | asymmetric         | 0.7481         | 0.2519     |

---

> > ### Author Rebuttal · Reviewer_4NSZ · 2026-04-07
> >
> > Thanks

---

> > > ### Author Response · Authors · 2026-04-07
> > >
> > > Thank you very much for your positive feedback and for your recognition of our work. We are pleased that our response was able to address all of your concerns.

---

### Official Review · Reviewer_DUZ2 · 2026-03-13

**Soundness:** 3
**Presentation:** 3
**Significance:** 2
**Originality:** 2
**Overall Recommendation:** 2
**Confidence:** 4

**Summary:**

This paper proposes Cerberus, a defense framework designed to protect collaborative perception (CP) systems used in connected and autonomous vehicles against adversarial attacks. Collaborative perception extends the perception range by sharing intermediate feature representations between vehicles, but this shared communication channel also introduces vulnerabilities, where malicious agents can inject manipulated features to disrupt perception outputs. The proposed method shifts the defense focus from traditional output-level consensus methods to feature-level consistency verification in the latent representation space.

**Compliance With Llm Reviewing Policy:**

Affirmed.

**Key Questions For Authors:**

1. How does the method behave if the ego vehicle’s feature representation is itself degraded or attacked?
2. Have the authors evaluated attacks that directly optimize against the joint anomaly score?
3. How does the computational overhead scale when the number of collaborating vehicles increases significantly?
4. How sensitive are the results to the balancing weights $\lambda_1$, $\lambda_2$, $\lambda_3$​ , and the uncertainty modulation parameter $\beta$ ?
5. The evaluation includes several fusion architectures. Could the authors clarify whether the method is truly architecture-agnostic or whether it requires tuning for different backbone networks?

**Limitations:**

Yes, the paper includes an Impact Statement that discusses potential dual-use concerns related to adversarial research and states that the experiments were conducted in controlled environments without physical risk.

**Strengths And Weaknesses:**

**Strengths:**

- The proposed defense framework is technically reasonable and is based on the observation that adversarial perturbations often create inconsistencies in feature representations. The paper clearly defines the three verification metrics, topological, semantic, and energy-based, and formulates them mathematically.

- The motivation for moving from output-level consensus to feature-level verification is clearly presented, and the architecture of Cerberus is well-explained.

- The concept of enforcing multi-dimensional feature-space consistency across collaborating agents is an interesting alternative to existing consensus-based or classifier-based defenses.

**Weaknesses:**

- The Cerberus framework's isomorphic view fusion assumes that surrounding vehicles are somewhat evenly distributed in a 360-degree radius to create equivalent virtual sensor arrays. In skewed real-world traffic situations, such as driving alongside a massive convoy or on an empty highway with cars only directly ahead or behind, interleaved sampling cannot produce truly equivalent fields of view.

- The framework's uncertainty-aware adaptive decision mechanism relies on a dynamic sliding window of historical anomaly scores to establish its normal baseline. A patient attacker could execute a "boiling the frog" strategy by continuously injecting microscopic, sub-threshold perturbations over hundreds of frames. Because these tiny deviations never trigger a rejection, they are added to the sliding window, gradually inflating the moving average and standard deviation until the threshold is high enough to allow a massive, targeted attack to slip through undetected.

- Although the framework introduces multiple metrics for feature consistency, the paper provides limited theoretical analysis to explain why these metrics can reliably distinguish benign and adversarial features across different models or environments.

- The defense relies heavily on the ego vehicle’s feature representation as the reference. If the ego perception system is itself compromised or significantly degraded, the detection reliability may decrease.

- It remains unclear how the defense performs against stronger adaptive adversaries that are specifically optimized against the proposed anomaly score.

- The evaluation mainly focuses on relatively small collaborative settings. It is still unclear how the computational overhead and verification procedure scale to larger multi-vehicle networks.

---

> ### Author Rebuttal · Authors · 2026-03-29
>
> [W1] For isomorphic view, we clarify that our interleaved sampling does not guarantee a perfect 360-degree view, but aims for relative spatial consistency between subgroups. In highly skewed traffic, our grouping still distributes the available views evenly. Consequently, while neither group achieves 360-degree coverage, their feature maps remain structurally comparable to each other where data exists. Cerberus relies entirely on this relative equivalence rather than absolute completeness. To validate this conclusion, we supplement the experiments in Table 1. The results show that, even under the front120 and front60 settings, where only vehicles within the forward 120° and 60° field of view are retained, Cerberus still achieves a high detection rate with a low false positive rate.
>
> *Table 1. Defense performance in skewed traffic scenarios*
> | Scenario | Attack   | Detection Rate (%) | False Positive Rate (%) |
> |-|-|-|-|
> | Front120 | MOR  | 91.30  |0|
> | Front60  | MOR  | 100.00  |0|
> | Front120 | TOR  | 91.30   | 0 |
> | Front60  | TOR  | 100.00  | 0 |
>
> [W2] We thank the reviewer for proposing this sophisticated threat model. However, executing this gradual poisoning strategy is practically infeasible against Cerberus for two reasons. First, our framework enforces multi-dimensional orthogonal verifications. Injecting microscopic perturbations to gradually inflate the energy score inevitably causes abrupt, easily detectable violations in structural consistency. Second, our adaptive mechanism incorporates a strict global maximum threshold as an absolute ceiling, which fundamentally prevents the sliding window from ever inflating enough to allow a massive attack to slip through undetected. Furthermore, we empirically implemented this attack: after a 5-frame warm-up to initialize statistics, we executed a 10-iteration poisoning phase  to gradually inflate the threshold, finally launching MOR and TOR attacks for 20 iterations. The Table 2 demonstrates our defense's robustness against it.
>
> *Table. 2 Robustness against persistent adversaries*
> | Model | MOR (ORR/ASR)  | TOR (ASR) |
> |-|-|-|
> | AttFusion | 4.55% / 0.04%  | 4.60% |
> | CoAlign | 2.18% / 0% | 0%|
> | Where2comm | 6.18% / 0.09% | 6.97%  |
> | V2VAM |10.85% / 0.39%| 20.52% |
>
> [W3] We will supplement more details of the theoretical analysis in the revised manuscript.
>
> [W4&Q1] Regarding the compromise and degradation of the ego vehicle, we respond from two aspects.
>
> *Compromised ego.* Similar to ROBOSAC (ICCV'23) and CP-Guard (AAAI'25), we assume that the ego vehicle is trustworthy, for two reasons. First, our defense focuses on protecting the ego vehicle as the fusion center against untrustworthy data transmitted from other collaborators. Second, if an attacker can directly compromise the ego vehicle, they can manipulate the final fused perception result without needing to craft adversarial transmissions from collaborators, rendering the collaborative attack vector unnecessary.
>
> *Degraded ego.* Our method incorporates an uncertainty-aware adaptive decision mechanism that dynamically adjusts decision thresholds in response to environmental variations, thereby mitigating the risk of false positives and reducing the impact of ego feature degradation, as validated in Fig. 4 of the paper.
>
> [W5&Q2] Section 5.4 (Table 4) evaluates our defense against a strong adaptive adversary with full knowledge of our defense mechanism, who directly optimizes the joint anomaly score. Under the adaptive attack, Cerberus reduces ASR to below 9.84% across all models and attack types, while preserving detection performance with mAP@0.5 up to 0.87 and mAP@0.7 up to 0.79. We also evaluate robustness under stronger attack parameters, please refer to our response to Reviewer 7TGc [Q1].
>
> [W6&Q3] Regarding computational overhead, the cost of our method scales almost linearly with the number of vehicles $N$. Specifically, sorting vehicles by azimuth requires $O(N \log N)$ complexity, while feature fusion requires $O(N)$. Crucially, the computation of anomaly scores operates on fused group level feature maps, making its complexity $O(1)$ with respect to $N$. Table 3 presents the computational overhead of our defense method, which demonstrates a significant performance advantage compared to the baselines.
>
> *Table 3. End-to-end defense latency  (ms)*
> | Method | N=5 | N=10 | N=15 | N=20 | N=30 | N=40 |
> | - | - | - | - | - | - | - |
> |Ours | 67.71 | 107.95 | 124.68 | 147.68 | 195.09 | 259.72 |
> | Robosac (ICCV'23) |112.61|193.51| 270.65 | 350.75 | 530.35 | 775.75 |
> | CP-Guard (AAAI'25)|231.35|572.76|799.85|1203.12|1715.92|2920.73|
>
> [Q4] Experiments and analyses have been supplemented. Due to word limits, please see our response [W4] to Reviewer 4NSZ.
>
> [Q5] Our method is architecture agnostic and does not require specific fine tuning for different models. Essentially, our method detects anomalies directly on feature maps across three dimensions, making it independent of the underlying model architecture.

---

> > ### Author Rebuttal · Reviewer_DUZ2 · 2026-04-04
> >
> > Thank you for the clarifications. The additional experiments are helpful and address some of my concerns; however, the response regarding theory remains limited.

---

> > > ### Author Response · Authors · 2026-04-06
> > >
> > > Thank you very much for your response and valuable comments. We sincerely thank the reviewer for acknowledging our additional experiments. We fully understand your remaining concern regarding the theoretical depth of our work. To address this concern, we have supplemented the theoretical analysis that was previously omitted due to space limitations. This analysis is presented as follows, and we hope it will resolve your questions.
> > >
> > >
> > > **Theorem 1 (Angular Equilibrium of Interleaved Allocation)** Let the azimuth angles of $N$ neighboring vehicles be arbitrarily distributed over $[-\pi, \pi]$, and $N_{\text{sorted}}$ be the sequence sorted by azimuth. Define the maximum adjacent angular difference as $\Delta_{\max} = \max_{1 \leq i < N} (\theta_{i+1} - \theta_{i}).$ After interleaving the vehicles into two groups $G_1$ and $G_2$, the angular difference between correspondingly positioned elements satisfies: $|\theta_{G_1}^{(k)} - \theta_{G_2}^{(k)}| \leq \Delta_{\max}.$ Specifically, when the $N$ vehicles are uniformly distributed over $[-\pi, \pi]$, the difference satisfies:$$|\theta_{G_1}^{(k)} - \theta_{G_2}^{(k)}| \leq \frac{2\pi}{N} \xrightarrow{N \to \infty} 0.$$
> > >
> > > **Theorem 2 (Upper Bound of the False Positive Rate for the Adaptive Threshold)**
> > > Assume the anomaly scores of benign vehicles within a statistical window follow a Gaussian distribution $J_k \sim \mathcal{N}(\mu_t, \sigma_t^2)$. Under the statistical threshold $T_{\text{stat}} = \mu_t + 3\sigma_t$, the one-sided false positive rate is: $P(\text{FP}) = P(J_k > T_{\text{stat}} \mid \text{benign}) = 1 - \Phi(3) \approx 0.00135$. After introducing uncertainty modulation, the final threshold becomes $\tau_t = T_{\text{stat}} + \beta U_t(T_{\text{max}} - T_{\text{stat}})$, at which point: $$P(\text{FP} \mid U_t) = 1 - \Phi\left(3 + \frac{\beta U_t(T_{\text{max}} - T_{\text{stat}})}{\sigma_t}\right)$$ Since $\Phi(\cdot)$ is monotonically increasing, and $U_t \geq 0$, $\beta > 0$, and $T_{\text{max}} > T_{\text{stat}}$, therefore: $$P(\text{FP} \mid U_t) \leq P(\text{FP} \mid U_t = 0) \approx 0.00135$$
> > > That is, the false positive rate of the adaptive threshold under any uncertainty conditions does not exceed the false positive rate of the fixed 3-sigma threshold, and it monotonically decreases as environmental uncertainty increases.
> > >
> > > **Theorem 3 (Compression of the Attacker's Feasible Region)**
> > > Let $F_{\text{topo}}$, $F_{\text{sem}}$, and $F_{\text{mag}}$ be the feasible perturbation sets for an attacker to evade topological, semantic, and magnitude detections, respectively.
> > > An attacker needs to simultaneously evade three detection dimensions, its feasible perturbation set is: $F_{\text{joint}} = \lbrace\delta : D_{\text{topo}}(\delta) < \tau_1\rbrace \cap \lbrace\delta : D_{\text{sem}}(\delta) < \tau_2\rbrace \cap \lbrace\delta : D_{\text{mag}}(\delta) < \tau_3\rbrace$. Since $D_{\text{topo}}$ constrains the structure of the features, $D_{\text{sem}}$ constrains the feature direction, and $D_{\text{mag}}$ constrains the energy magnitude, the three act on different subspaces of the feature space, therefore: $F_{\text{joint}} \subseteq F_{\text{topo}} \cap F_{\text{sem}} \cap F_{\text{mag}}$
> > > Then, the measure of $F_{\text{joint}}$ satisfies:
> > > $$\mu(\mathcal{F}_{\text{joint}}) \leq \min_k \mu(\mathcal{F}_k).$$
> > > Consequently, the bypassable region of joint detection is no greater than the bypassable region of any single-dimensional detection.
> > >
> > >
> > > Thank you again for your valuable comments. We hope that our responses have addressed your concerns. If so, we would greatly appreciate it if you could reconsider your evaluation of our paper based on the revised responses.

---

### Official Review · Reviewer_nvAo · 2026-03-21

**Soundness:** 3
**Presentation:** 3
**Significance:** 3
**Originality:** 3
**Overall Recommendation:** 5
**Confidence:** 4

**Summary:**

This work proposes Cerberus, a multi-head defense strategy for detecting adversarial attacks in collaborative perception (CP). The method is capable of identifying whether malicious vehicles exist among connected and autonomous vehicles (CAVs), and can further localize the malicious participants. Specifically, Cerberus jointly analyzes the feature maps shared by neighboring vehicles and received by the ego vehicle from three perspectives: topological integrity, semantic direction, and energy distribution. It determines whether any of these feature maps have been adversarially manipulated. By aggregating these multi-dimensional consistency signals through weighted summation and applying dynamically adjusted thresholds, the method ultimately decides whether attackers and malicious feature representations are present.

**Compliance With Llm Reviewing Policy:**

Affirmed.

**Key Questions For Authors:**

1. In the final stage of identifying malicious vehicles, the method adopts a process similar to binary search to progressively narrow down the candidate set and eventually localize specific vehicles. This implies that in the last round of grouping, each group contains only one or two vehicles. When the number of vehicles in each group becomes very small, can the overall pipeline still reliably detect discrepancies? If yes, why not directly compare each vehicle individually with the ego vehicle from the beginning? If not, how do the final rounds of grouping ensure reliable localization of malicious vehicles?
2. How does the method perform when there is more than one malicious vehicle? Can the output structure of the algorithm report multiple malicious vehicles simultaneously?
3. The Isomorphic View Fusion module is designed to address pseudo-inconsistencies caused by viewpoint differences. However, this issue is not limited to attack detection, it also exists in the normal collaborative perception (CP) data fusion process. Why is the solution to this problem introduced in this defense framework, rather than being addressed in the original CP methods or in separate prior work specifically targeting this issue?
4. In the first step of Cerberus, the ego vehicle groups the received feature maps and then evaluates each group independently for the presence of malicious features. However, during this process, there is no direct comparison between groups; instead, each group is compared against a threshold. If inter-group differences are not explicitly utilized, what is the purpose of grouping? Why not directly fuse all received feature maps into a single representation and determine whether it is malicious? This alternative could further reduce computational cost, as only one matrix needs to be evaluated, and attacks are relatively rare events.
5. Alternatively, if the purpose of Isomorphic View Fusion grouping is to increase the proportion of anomalous feature maps within each group, then the effectiveness of the method depends on this proportion falling within a certain range. In that case, grouping should not be performed via a simple binary split each time. Instead, group sizes should be controlled to maintain an effective anomaly ratio. For example, if the anomaly ratio must not fall below 10%, then each group should contain no more than 10 vehicles. If there are 50 vehicles participating in CP, it would be more appropriate to divide them into 5 groups of 10 rather than 2 groups of 25, since the latter would dilute the signal. Therefore, is the binary grouping strategy used in Isomorphic View Fusion truly optimal, or does it have specific advantages that alternative grouping strategies lack?
6. The x axis in Figure 5 are incomplete, the 10s are missing their power number.

**Limitations:**

Yes

**Strengths And Weaknesses:**

Strengths:
Existing approaches typically rely on output-level analysis, which is inherently vulnerable to manipulation, or employ machine learning models to learn patterns of malicious behavior, leading to limited generalization capability. In contrast, the proposed Cerberus defense framework focuses on Feature-Level Consistency Verification, which is intrinsically more difficult to forge. Moreover, it analyzes feature representations from three independent perspectives: topological integrity, semantic direction, and energy distribution, thereby enhancing both robustness and generalization. Experimental results demonstrate that Cerberus achieves strong performance against existing adversarial attacks, validating its feasibility and effectiveness.

Weaknesses:
The proposed Feature-Level Consistency Verification attempts to identify malicious feature maps from three perspectives: topological integrity, semantic direction, and energy distribution shift. However, all three criteria are largely heuristic, derived from observations of relatively basic existing attack strategies. While effective under current threat models, they may still be bypassed by more adaptive or sophisticated future attacks.

For topological integrity, the paper argues that objects such as road boundaries or bounding boxes evolve continuously over time and space, rather than appearing abruptly. However, this assumption can be circumvented. An attacker could introduce a fake object gradually (e.g., emerging from occlusion regions such as buildings or corners), or ensure that the fake object is consistently present from the beginning, thereby preserving apparent spatial continuity.

For energy distribution shift, the detection relies on identifying abnormal activation magnitudes. This constraint is relatively easy to evade, as an attacker could normalize or rescale the feature maps prior to transmission, aligning their energy statistics with those of benign features.

For semantic direction, the limitation is more fundamental. A core objective of collaborative perception is to detect objects outside the ego vehicle’s field of view. Consequently, the ego vehicle inherently lacks direct evidence to verify such objects. If an attacker places a fake target in a region that is invisible to the ego vehicle but appears plausible from the collaborator’s perspective, semantic inconsistency becomes difficult to establish. In such cases, the ego vehicle cannot reliably refute the injected semantics.

Overall, none of the three criteria are fundamentally unbypassable. Instead, they function as complementary constraints that increase the difficulty for an attacker to simultaneously satisfy multiple forms of “naturalness.” Therefore, the primary contribution of this work is not the complete resolution of adversarial attacks in collaborative perception, but rather a substantial increase in the stealthiness requirements and attack cost across multiple dimensions.

---

> ### Author Rebuttal · Authors · 2026-03-29
>
> [W] Regarding the weaknesses, we agree that each of the three criteria is not foolproof when considered in isolation. As accurately pointed out, a sophisticated attacker might attempt to bypass energy constraints via rescaling, or evade topological checks via gradual introduction. However, as you summarized, our true contribution lies precisely in utilizing them as complementary constraints. Enforcing these jointly creates an intractable multi-dimensional optimization trade-off, substantially raising the attack cost. Furthermore, our responses to Reviewer DUZ2 [W2] and Reviewer 7TGc [Q1] demonstrate our method's defensive capabilities against both stealthy persistent attacks and white box adaptive attacks, proving this conclusion from an experimental perspective.
>
> [Q1] Yes, the pipeline remains highly reliable at the final 1-vs-ego stage. If we performed individual 1-vs-ego comparisons, the system would require $N$ verification operations per frame, yielding an $O(N)$ complexity. Conversely, our approach executes a global group-level check first (1 operation). Only if an anomaly is triggered do we initiate the binary search, localizing the attacker in $O(\log N)$ operations. The table below compares the average evaluations per frame and average single vehicle checks per frame between our grouping and direct comparison. Our grouping approach significantly reduces these numbers, demonstrating the necessity of this strategy for minimizing verification overhead.
>
> *Table 1. Verification count: our grouping vs. per-vehicle*
> | Setting             | Mode             | Avg. Verifications per Frame | Avg. Single-Vehicle Verifications per Frame |
> |---------------------|------------------|------------------------|----------------------|
> | No Attack | Our Grouping    | 2.0                | 0               |
> | No Attack| Direct Per-Vehicle | 5.63                 | 5.63             |
> | MOR      | Our Grouping     | 4.47                | 2.04              |
> | MOR      | Direct Per-Vehicle | 5.63                | 5.63              |
> | TOR | Our Grouping    | 4.47                | 2.04              |
> | TOR | Direct Per-Vehicle | 5.63                | 5.63             |
>
> [Q2] Our method inherently supports defending against multiple malicious vehicles. Cerberus initially divides collaborators into two groups; any group containing attackers will yield a high anomaly score. This triggers recursive binary splits until all malicious vehicles are successfully localized. Fig. 9 in the manuscript also demonstrates that the proposed method maintains strong defensive performance in the presence of multiple malicious vehicles, further confirming its capability to defend against multiple adversarial vehicles.
>
> [Q3] Standard CP and our defense framework have fundamentally different objectives. Existing CP methods handle viewpoint differences implicitly (e.g., via max-pooling or attention) to maximize perception accuracy. However, these operations entangle features and destroy source-specific spatial details. Conversely, our defense requires explicit structural alignment to perform precise feature-level consistency checks. Therefore, the Isomorphic View Fusion module is uniquely designed to create structurally comparable subsets for adversarial verification, a strict security requirement that standard perception-driven fusion inherently does not address.
>
> [Q4] The grouping strategy serves two main purposes. First, it aims to divide vehicles into two statistically equivalent groups in terms of their viewpoint distribution relative to the ego, ensuring balanced viewpoint coverage across both group vs. ego tests and reducing false positives caused by grouping bias. Second, it reduces the number of verification rounds required. Fusing all vehicles only provides a binary attack/no attack alert. To actually localize the malicious vehicle, the system would still have to subsequently split the vehicles and perform additional verifications. Initial grouping preemptively halves the search space, fundamentally reducing the total verification steps required for localization during an attack.
>
> [Q5] The purpose of grouping is not to maximize the anomaly ratio within each group, but to construct two validation groups that are statistically equivalent in their angular/coverage distribution relative to the ego. This is achieved by sorting elements by relative angle and assigning them in an interleaved manner, so that each group is comparable to the ego vehicle rather than being directly compared with the other group. In addition, increasing the number of groups would further increase the validation overhead.
>
> [Q6] We will revise this in the updated manuscript to make the exponent clearer.

---

> > ### Author Rebuttal · Reviewer_nvAo · 2026-04-01
> >
> > Thank you for the detailed and thoughtful responses. I find that the clarifications provided for Q1–Q3 are satisfactory and have addressed our concerns. In particular, the computational advantages of the binary search-based localization strategy, as well as reason for re-inventing the method for handling viewpoint difference, are clear and well-justified.
> >
> > However, I would like to further follow up on Q4–Q5, as I believe the current response does not fully address the core of our concern.
> >
> > Our question is not about the role of grouping in attack-time localization, but rather about its necessity in the **initial validation stage under the no-attack regime**, which is expected to dominate in practical deployments. Specifically, the current pipeline performs two group-level validations per frame from the outset. In contrast, an alternative design would be to first fuse all received feature maps into a single representation and perform a single global validation. Only if an anomaly is detected would the system proceed to grouping and recursive localization. Such a design could significantly reduce the baseline computational overhead during normal operation.
> >
> > Therefore, the central question is:
> >
> > **What is the fundamental advantage of splitting into two groups at the initial stage, compared to performing a single global validation followed by conditional grouping only when needed?**
> >
> > In the current response, the authors emphasize that grouping ensures statistically equivalent viewpoint distributions and reduces the search space during localization. While these points are valid for attack-time behaviour, they do not directly explain why two groups are necessary before any anomaly is detected, especially given that no inter-group comparison is performed and each group is evaluated independently against the ego feature.
> >
> > This leads to a related concern:
> >
> > Since grouping does not introduce explicit inter-group signals, its role in the initial stage appears to be closer to redundant sampling rather than enhancing discriminative power. It remains unclear whether this redundancy is essential for maintaining detection sensitivity, or whether a single global validation would suffice as a first-stage filter.
> >
> > I would like to invite the authors to further clarify:
> >
> > 1. Whether a single global validation step would lead to a significant drop in detection recall (e.g., due to anomaly dilution), and if so, please provide empirical or theoretical justification.
> >
> > 2. Why the proposed design prioritizes immediate grouping over a potentially more efficient “coarse-to-fine” (lazy verification) strategy.
> >
> > 3. Whether the current grouping strategy offers intrinsic advantages in detection capability, beyond its role in reducing localization complexity.
> >
> > I believe clarifying this point would further strengthen the methodological justification of the proposed framework, particularly in terms of its efficiency and practical deployment considerations.

---

> > > ### Author Response · Authors · 2026-04-05
> > >
> > > Thank you very much for your detailed response. We also sincerely apologize for not clearly addressing your concerns in our previous response. The suggestion of a coarse-to-fine (lazy verification) strategy is intuitive and elegant from a system efficiency standpoint. However, we clarify that our initial grouping is fundamentally driven by the need to preserve detection sensitivity, which a single global validation inherently compromises due to the anomaly dilution effect. Specifically, when the attack strength is low, the collaborative perception fusion process may suppress the malicious features, thereby leading to a decrease in detection recall.
> > >
> > > As shown in Table 1, we report the detection recall under PGD attacks with different strengths when the number of collaborating vehicles is 11, comparing our grouping method with the lazy verification approach. The results show that as the attack strength decreases, the lazy verification approach tends to miss attacks. In contrast, our grouping method can consistently detect the attacks. This demonstrates the necessity of our grouping strategy. Next, we will respond in detail to the three new questions you raised.
> > >
> > > *Table1. Comparison between our grouping strategy and lazy verification under different attack intensities $\epsilon$*
> > > | Grouping Strategy |$\epsilon $ | Detection Recall |
> > > |---|---|---|
> > > | Ours | 0.030 | 80.43% |
> > > | Lazy Verification | 0.030 | 0.00% |
> > > | Ours | 0.032 | 80.43% |
> > > | Lazy Verification | 0.032 | 0.00% |
> > > | Ours | 0.034 | 80.43% |
> > > | Lazy Verification | 0.034 | 0.00% |
> > > | Ours | 0.036 | 80.43% |
> > > | Lazy Verification | 0.036 | 0.00% |
> > > | Ours | 0.038 | 80.43% |
> > > | Lazy Verification | 0.038 | 2.17% |
> > > | Ours | 0.040 | 80.43% |
> > > | Lazy Verification | 0.040 | 10.87% |
> > > | Ours | 0.042 | 80.43% |
> > > | Lazy Verification | 0.042 | 26.09% |
> > > | Ours | 0.044 | 80.43% |
> > > | Lazy Verification | 0.044 | 39.13% |
> > > | Ours | 0.045 | 80.43% |
> > > | Lazy Verification | 0.045 | 52.17% |
> > >
> > > **Whether single global validation leads to a decrease in detection recall**
> > >
> > > Yes, single global validation leads to a decrease in detection recall. In a single global validation, one malicious feature map is fused alongside N-1 benign maps. Due to the nature of standard CP fusion operators (e.g., max-pooling or attention), the malicious spatial activations are severely smoothed and suppressed by the overwhelming benign context. Consequently, the overall anomaly score drops below the detection threshold, causing the coarse first-stage filter to fail entirely.
> > >
> > > By splitting the traffic into two groups at the outset, we strictly limit the denominator of this dilution. The malicious feature is mixed with only N/2-1 benign features, effectively doubling its anomaly concentration within its specific subgroup. Therefore, the initial grouping is not merely redundant sampling, it is a structural necessity to maintain the discriminative power required to trigger the alarm in the first place.
> > >
> > > In particular, the results in Table 1 show that, compared with our grouping strategy, this lazy strategy, which first performs global validation and only applies grouping after detecting anomalies, exhibits a certain decrease in detection recall.
> > >
> > > **Reasons for prioritizing immediate grouping over lazy verification**
> > >
> > > We prioritize immediate grouping because security systems cannot trade critical detection recall for marginal efficiency gains during the initial validation stage. Although lazy verification may appear more efficient, it suffers from a decrease in detection recall. In contrast, the grouping strategy reduces the number of vehicles to be validated and constructs two homogeneous groups of vehicles for validation, which enables our method to detect attacks effectively. Moreover, our method also incurs relatively low time overhead and shows significant advantages over other defense methods.
> > >
> > > **Inherent advantages of the grouping strategy**
> > >
> > > Besides reducing localization complexity, the intrinsic advantage of the grouping strategy lies in preventing anomaly dilution. As shown in Table 1, it effectively mitigates the anomaly dilution caused by the first-round global validation and helps ensure the stable operation of the security mechanism. In particular, when the perturbation intensity is below 0.038, our grouping strategy can still detect the attack. These experimental results further demonstrate the advantage of this strategy.
> > >
> > > Thank you again for your prompt response and for your more in-depth comments on our manuscript. We will incorporate all of these suggestions into the revised version.

---

### Decision · Program_Chairs · 2026-04-30

**Decision:**

Accept (regular)

**Comment:**

This paper proposes Cerberus, a defense framework for adversarial robustness in collaborative perception that shifts from output-level consensus to feature-level consistency verification across multiple dimensions. Reviewers agree that the problem is important and timely, and find the core idea of leveraging multi-dimensional feature-space consistency both intuitive and technically meaningful. The method is well-motivated, clearly presented, and supported by strong empirical results demonstrating substantial improvements in attack mitigation and perception recovery. While some reviewers note that individual consistency criteria are heuristic and may be bypassed in isolation, there is broad agreement that their joint use significantly increases the difficulty for adaptive attackers, effectively raising the attack cost. The rebuttal further strengthens the paper by providing additional experiments, clarifications, and theoretical discussion, addressing most of the concerns regarding robustness, scalability, and practical deployment. Overall, the paper offers a solid and practically relevant contribution to secure collaborative perception, with strong empirical validation and a clear design rationale.